# Turning main-group element magnesium into a highly active electrocatalyst for oxygen reduction reaction

Shuai Liu [1], Zedong Li[1], Changlai Wang [1], Weiwei Tao[3], Minxue Huang[1], Ming Zuo[1], Yang Yang[1], Kang Yang[1], Lijuan Zhang[4], Shi Chen [1], Pengping Xu[1] & Qianwang Chen[1,2] ✉

It is known that the main-group metals and their related materials show poor catalytic activity due to a broadened single resonance derived from the interaction of valence orbitals of adsorbates with the broad sp-band of main-group metals. However, Mg cofactors existing in enzymes are extremely active in biochemical reactions. Our density function theory calculations reveal that the catalytic activity of the main-group metals (Mg, Al and Ca) in oxygen reduction reaction is severely hampered by the tight-bonding of active centers with hydroxyl group intermediate, while the Mg atom coordinated to two nitrogen atoms has the near-optimal adsorption strength with intermediate oxygen species by the rise of p-band center position compared to other coordination environments. We experimentally demonstrate that the atomically dispersed Mg cofactors incorporated within graphene framework exhibits a strikingly high half-wave potential of 910 mV in alkaline media, turning a s/p-band metal into a highly active electrocatalyst.

---

[1] Hefei National Laboratory for Physical Science at Microscale and Department of Materials Science & Engineering, University of Science and Technology of China, Hefei 230026, China. [2] Anhui Province Key Laboratory of Condensed Matter Physics at Extreme Conditions, High Magnetic Field Laboratory of Chinese, Academy of Sciences, Hefei 230031, China. [3] Department of Mechanical Engineering, Boston University, Boston, MA 02215, USA. [4] Shanghai Synchrotron Radiation Facility, Shanghai Institute of Applied Physics, Shanghai 201203, China. ✉email: cqw@ustc.edu.cn

Electrochemical energy conversion is promising for meeting growing energy demand and to reduce environmental pollution from traditional catalysts[1,2], thus electrocatalysis is a topic that is gaining interest. High-performance and cost-efficient electrocatalysts are of great importance to energy conversion processes. The development of renewable energy technologies, such as polymer electrolyte membrane fuel cell (PEMFC) and metal-air batteries via the oxygen reduction reaction (ORR), are heavily dependent on the improvement of highly active electrocatalysts[3].

In the past decades, the d-band center model proposed by Hammer and Nørskov has been widely applied in understanding and predicting catalytic activity on transition metal-based electrocatalysts[4–6]. A single parameter, the center position of d band ($\varepsilon_d$), is linearly related to the adsorption free energies of adsorbates on transition metal surface[7]. An optimal d-band structure with moderate adsorption strength is essential in catalyst design, which can be demonstrated by the volcano-type activity plots[8,9]. More specifically, the interaction between the adsorbate states and the metal d-states gives rise to a deep-lying filled bonding state and a partially filled anti-bonding state. The adsorption strength is given by the filling of the anti-bonding state, a neither too strong nor too weak bonding is in favor of catalysis[10,11]. For a more in-depth description, the lower $\varepsilon_d$ down from Fermi level is, the lower resultant anti-bonding level will be (increased occupancy), which leads to a weaker adsorption[12]. For instance, Stamenkovic et al. realized a high ORR catalytic activity for $Pt_3Ni$ alloy by turning the electronic structure of surface Pt atoms into a lower d-band center position[13]. With increased occupancy of anti-bonding state of OH*, weaker adsorption is induced on $Pt_3Ni$ surface ($\Delta G_{OH*}$, which is closer to volcano-type center), therefore the activity of $Pt_3Ni$ becomes higher than pure Pt catalyst.

On the contrary, main group metals, such as Mg and Al are thought to be catalytic inactive[14] due to lacking the combination of empty and filled host-orbitals that is crucial for electronic processes involved in elementary steps in catalytic cycles[15], especially for ORR process involving multi oxygenated intermediates transformation. However, no one has ever set foot into this virgin land.

Different from a narrow d-band for transition metals, main group metals sites with a delocalized s/p-band as the host-orbital mainly broadens the adsorbate state due to "weak chemsorption"[7,11]. The interaction of adsorbates guest-states with those unbefitting surface-states results in too strong adsorption (deep and filled resonance state makes adsorption site poisoned) or too weak adsorption (makes no activation of adsorbates). To the best of our knowledge, main group metals serving as active catalytic sites during electrocatalytic reaction has not been reported so far. However, in nature, enzymes containing the magnesium (Mg) cofactors have key roles in many vital metabolic pathways and nucleic acid biochemistry[16], for example, Mg-isocitrate in isocitrate lyase[17]. Typically, $Mg^{2+}$ in cofactor has a suitable affinity for oxygenated species, so the cofactor can function as a Lewis acid for the activation of Mg-bound water to a hydroxide ion or transfer of a phosphate intermediate from one compound to another[16]. For instance, $Mg^{2+}$-bound DNA and RNA polymerases are efficient as they participate in neutralization of the polyanionic charge of the nucleic acid for especially suitable affinity to oxygen atoms[18,19].

The catalytic activity for ORR is strongly related to the adsorption strength of oxygen-bearing intermediates (O*, OH*, and OOH*) at catalytic sites[8,20]. As mentioned above, the Mg cofactor with suitable adsorption strength to oxygenated species could be favorable for ORR if its p-electronic state is tuned to a reasonable level. Direct mimicking the Mg cofactors

configuration to design catalysts is not applicable due to their instability in experimental environment when the complex protein-based substrate does not exist[21]. Indeed, the intrinsic activity of catalytic sites is strongly correlated with their electronic states which can be tuned by altering the coordination number of the sites[22]. The previous theoretical calculation shows that surface p-state of Mg atom can be altered by nitrogen-heterocyclic molecules like porphyrins via Mg–N covalent bonding[23–25]. This interaction can change p-orbital electrons occupancy of Mg site, which shifts the energy levels of the highest occupied molecular orbital (HOMO)[25,26]. Thus, a reasonable p-band energy level of Mg atom can be tuned by changing coordination environments, making Mg site active to catalyze oxygenated species transformation in ORR[27].

In this work, graphene-based N-coordinated metal cofactors centered with main group metals, Mg, Ca, and Al were investigated by both density function theory (DFT) simulations and experimental studies. The variation trends of the p-band centers of the main group metals cofactors and key intermediates adsorption strength on metals sites are well established. It is proved that the p-band center of Mg shifts upward, the adsorption strength of oxygenated species decrease, which is contrary to the trend observed in d-band metals. It indicates that the catalytic activity of main group metal Mg can be tuned by coordination with two N atoms changing its local N-coordination, which derived from the modulation of p-band filling and then influences ORR catalytic activity. The single Mg atom catalysts shows extremely high ORR activity, which surpasses the performance of commercial Pt/C in alkaline conditions and far exceeds that of most transition metal-based catalysts reported so far.

## Results

**Calculation of electrocatalytic activity**. DFT simulations were performed using Vienna Ab Initio Simulation Package (VASP)[28,29] to study the possible catalytic activity and reaction mechanism in ORR. Three types of metal atoms (Mg, Ca, and Al) coordinated with different numbers (from 1 to 4) of pyridine nitrogen atoms are confined in a graphene matrix (Fig. 1a and Supplementary Fig. 1) which is denoted as M (C or A) $N_xC$ according to N coordination numbers. Alkaline electrolyte, commonly used in a typical ORR experiment for N-coordinated metal moieties, is chosen to evaluate the ORR performance[10,30]. The free energy diagram for ideal ORR catalyst model is illustrated in Supplementary Fig. 2. The equilibrium potential is set to be 1.23 V versus reversible hydrogen electrode (RHE) (according to Nernst equation, for each step in ORR process with 4.92 V/4 = 1.23 V vs. RHE), where the reaction free energies for all electron-transfer steps are zero for the ideal model when onset potential equals to equilibrium potential.

As a benchmark, the free energy diagram of commercial Pt/C is also calculated using the Pt (111) model, which is a widely used theoretical model for Pt. As shown in Supplementary Fig. 3, when the output potential is set to zero relative to RHE, ORR happens spontaneously for Pt (111) since all electron-transfer steps are exergonic. However, when the output potential increases over 0.83 V, the free energy of one of the electron-transfer step (the last step here) becomes positive, while other steps remain negative, which indicates that the onset potential for Pt (111) is 0.83 V.

The onset potential ($U_{RHE}^{onset}$), output potential for starting of spontaneous ORR process, is determined by the reaction free energy of each electron-transfer step or adsorbate strengths of the reaction intermediates (OOH*, O*, OH*). The optimized configurations for adsorbates at metal cofactors (OOH*, O*, and OH*) are given in Supplementary Figs. 4–6. The free energy diagram for all models with output potential of 0, 0.83, and 1.23 V are drawn in Supplementary Figs. 7–9. It reveals that $O_2$

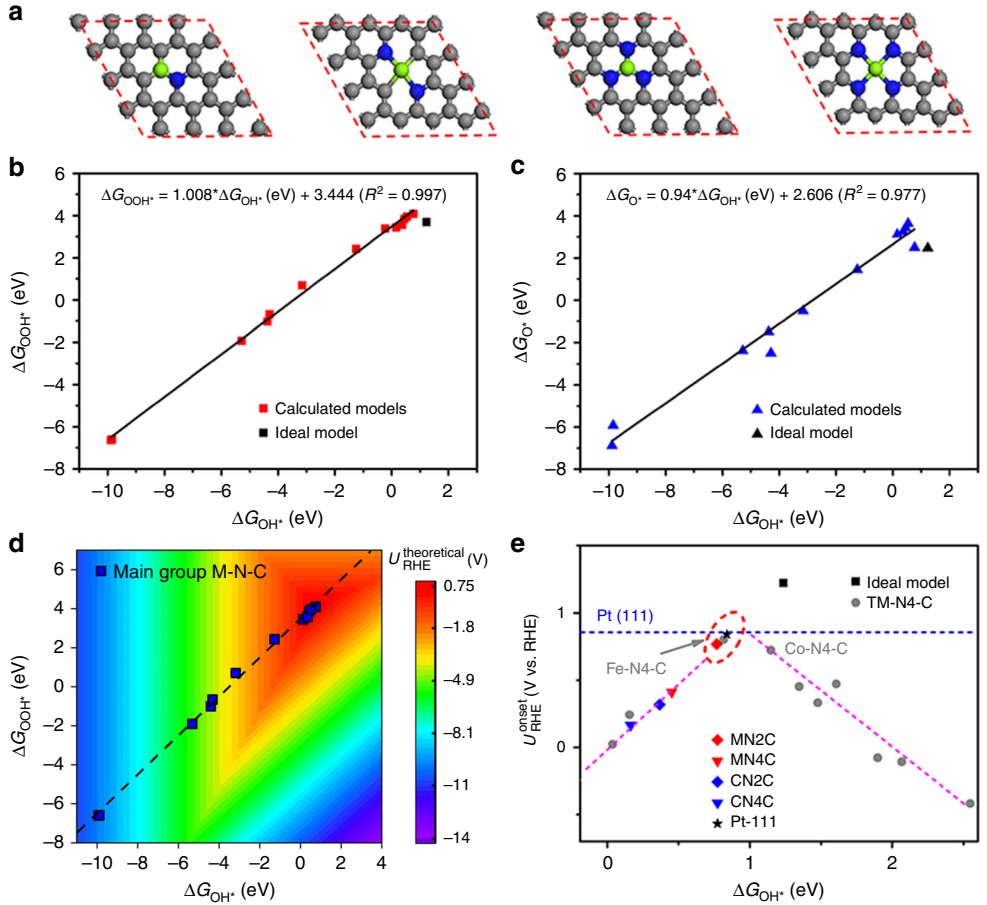

**Fig. 1 The calculation models and results. a** The geometric configurations of Mg embedded in graphene framework predominantly existing as single atoms coordinated with different numbers (from 1 to 4) of nitrogen atoms. The green, blue, and gray balls represent metal, N and C atoms, respectively. **b** Adsorption free energies of OOH as a function of that of OH for models of Mg, Al, and Ca coordinated with different numbers of pyridine nitrogen atoms embedded in graphene matrix. **c** Adsorption free energies of O versus that of OH for all models. **d** The two-dimensional volcano map about theoretical onset potential versus adsorption free energies $\Delta G_{OH^*}$ and $\Delta G_{OOH^*}$ on main group metal cofactors (M-N-C). **e** The zoomed-in view of the onset potential versus $\Delta G_{OH^*}$ in the region close to the performance ceiling of models and transition metal cofactors (TM-N4-C).

transform to OOH radical at Mg and Al sites are exothermic but OH* adsorption strength at Al and Ca sites are too stronger than Mg sites and thus induce large energy barrier in elementary steps, so only Mg sites show better ORR performance.

**Adsorption–activity relationships**. According to previous definition[28], the ORR electrocatalytic performance can be predicted by intermediates adsorption energy. Therefore, it's important to identify their relationship so as to help the design of electrocatalysts. The reaction free energies for each step and the adsorption free energies of adsorbates for all three different metal-centered catalysts with various N neighbors are listed in Supplementary Tables 1–3. Figure 1b, c plot the absorption free energy for OOH* and O* as a function of that for OH*. All data have a linear relationship that can be fitted by:

$$\Delta G_{OOH^*} = 1.008 \times \Delta G_{OH^*} + 3.444 \left(R^2 = 0.997\right)$$

$$\Delta G_{O^*} = 0.94 \times \Delta G_{OH^*} + 2.606 \left(R^2 = 0.977\right)$$

which is consistent with reported linear relationship existing in transition metal based catalysts[20,31,32]. According to those equations, $\Delta G_{OH^*}$ is chosen to be the only independent variable to describe $U_{RHE}^{onset}$ in all models[33].

The onset potentials for each model are listed in Supplementary Tables 4-5. Figure 1d plots the $U_{RHE}^{onset}$ volcano map as a

function of $\Delta G_{OH^*}$ and $\Delta G_{OOH^*}$. The $U_{RHE}^{onset}$ for all Mg, Al, and Ca-centered catalysts linear located at the left side of the volcano map where $\Delta G_{OH^*}$ is less than 1 eV. It indicates that the badly strong binding of OH* severely hampers their ORR process as the unbefitting host-levels of most metal sites. Thus $U_{RHE}^{onset}$ of most models are far away from the volcanic map top, in other words, they show bad ORR performance. However, a few dots located near the volcano map top and Fig. 1e shows the zoomed-in view where $\Delta G_{OH^*}$ ranges from 0 to 2 V. For comparison, the data points of transition metal–nitrogen–carbon (TM–N–C) models are plotted according to values reported by Xu et al.[10]. Different from both sides location for TM–N4–C models, main group metal-centered models are located at left side indicating stronger hydroxyl adsorption strength. However, interestingly, the data point of M2NC model is located near the top of the volcano map. The onset potential of MN2C is 0.77 V, which is comparable to the $U_{RHE}^{onset}$ of Pt (111) (0.83 V) under alkaline conditions and much higher than that of Al and Ca cofactors. The MN2C with moderate oxygenated species adsorption strength close to volcano top may be catalytic active.

**Relationship between adsorbate strength and p-state feature**. Figure 1 illustrates that the ORR performance of a metal center changes with different local coordination environments, e.g., MN2C exhibits higher ORR activity compared to that of MN3C

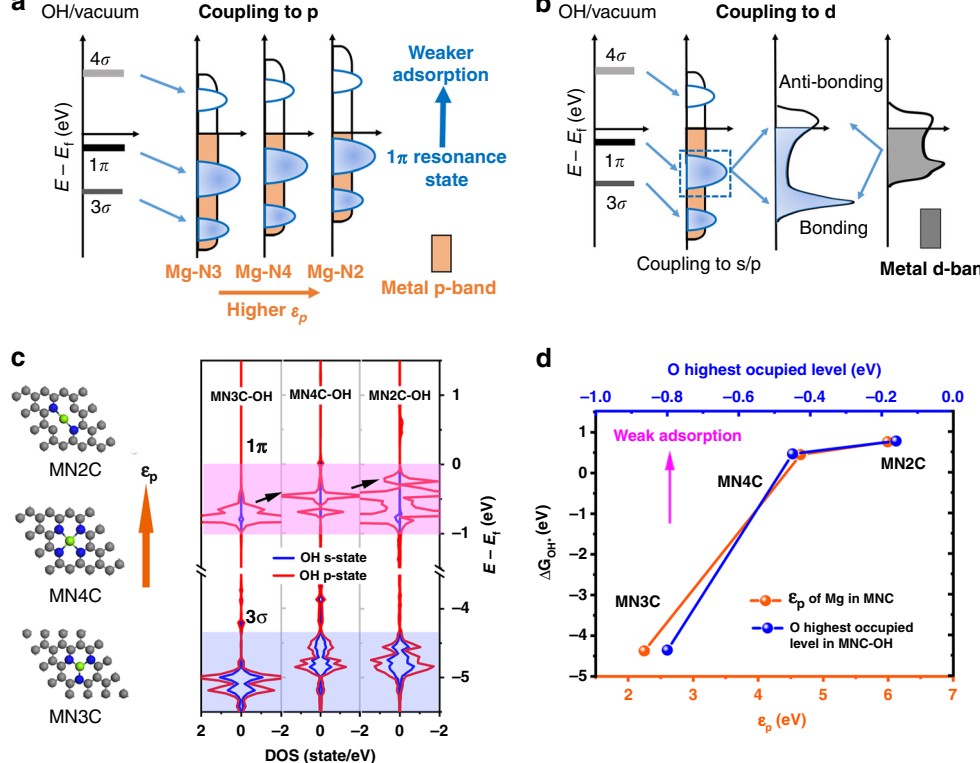

**Fig. 2 OH\* Adsorption-energy-based activity descriptors for p-band Mg cofactors. a, b** Schematic illustration of the change in local electronic structure at OH molecular orbitals upon adsorption on the surface of p-band metal or d-band metal. **c** The Mg cofactors' geometries in different N-coordinated environments which tune $\varepsilon_p$ of Mg atoms, and the corresponding density of states changes of OH after interaction with p-state of Mg. **d** Adsorption free energies of OH as a function of the $\varepsilon_p$ position of metal atoms for Mg cofactors, and also as a function of highest O occupied state of hydroxyl after interaction. Lines are used to fit the linear relationship.

and MN4C due to the near optimal adsorption strength with intermediate OH species. However, the mechanism is unknown. In order to understand the change of OH\* adsorbate strength with different coordination environment, density of states (DOS) for MNxC models are examined (x from 2 to 4 and MN1C is excluded since it is not stable as shown in Supplementary Fig. 4).

It should be noted that there is difference in adsorbate-surface interaction when adsorbate interacts with a broad s/p-band or a narrow d-band. The Newns–Anderson–Grimley model can well describe this change[34,35]. As illustrated in Fig. 2a, when OH vacuum states interact with broad p-band, they are shifted down due to the interaction and broadened into resonances. The $1\pi$ (from lone pair O $2p_{x, y}$ electrons) and $3\sigma$ (from H $1s$-O $2p_z$)[36,37] are renormalized. The energy level near the Fermi level is responsible to the binding strength. With higher p-band location (higher p-band center position ($\varepsilon_p$)), take Mg–N2 as example, the $1\pi$ resonance state after interaction is closer to Fermi level, which results in weaker adsorption strength. In comparison, as illustrated in Fig. 2b when interacts with d-band metals, the OH vacuum states are first down shifted and broadened by s/p state, whereas subsequent coupling to d-state gives rise to the splitting of state. The binding strength is correlated with the filling of anti-bonding state near Fermi surface.

The model is helpful to understand the adsorption difference. Here, it is found that the coordination environment changes the electrons filling of 3p orbital of Mg centers after hybridization. The combination of partially filled and extra empty p-orbitals is described as p-band center position ($\varepsilon_p$) of Mg sites (see Supplementary Table 6 and Supplementary Fig. 10). As shown in Fig. 2c, the MN2C processes the highest $\varepsilon_p$ compared to

others., after interaction with Mg site, the p-adsorbate resonance state at OH molecular (especially $1\pi$) in local DOS is higher (as shown in Fig. 2c with black arrows), which means weak adsorption strength. This trend is tabulated in Fig. 2d. The MN2C with the highest $\varepsilon_p$ of Mg atom processes the weakest OH adsorption than others (as shown with orange dots), which is due to the highest interacted $1\pi$ occupied state (as shown with blue dots).

This trend is also found in Al and Ca cofactors as shown in Supplementary Fig. 11. So MN2C with suitable p-state tuned by coordination can also bring similar adsorption strength like d-band metals.

**The synthesis of graphene-based N-coordinated Mg cofactors.** In addition, Fig. 3a shows the preparation scheme of graphene-based Mg-centered cofactor catalyst Mg–N–C. It is prepared by direct pyrolysis of Mg-based metal-organic-framework (MOF) (named as Mg-HMT) and then acid pickling. Most of Mg is evaporated after pyrolysis except for some of them are bonded with surrounding atoms in graphene carbon matrix (see Supplementary Information 1.2 part). Acid pickling is performed to get rid of possible impurity and finally the Mg–N–C is synthesized. The details are discussed in section 1.2 in Supplementary Information.

As shown in Supplementary Fig. 17d, e, the X-ray diffraction pattern (XRD) and the Raman spectra reveal that the catalyst is composed of defect-rich graphitic carbon. The high resolution transmission electron microscopy (HRTEM) image of Mg–N–C in Fig. 3b confirms that Mg–N–C is the porous graphitic carbon.

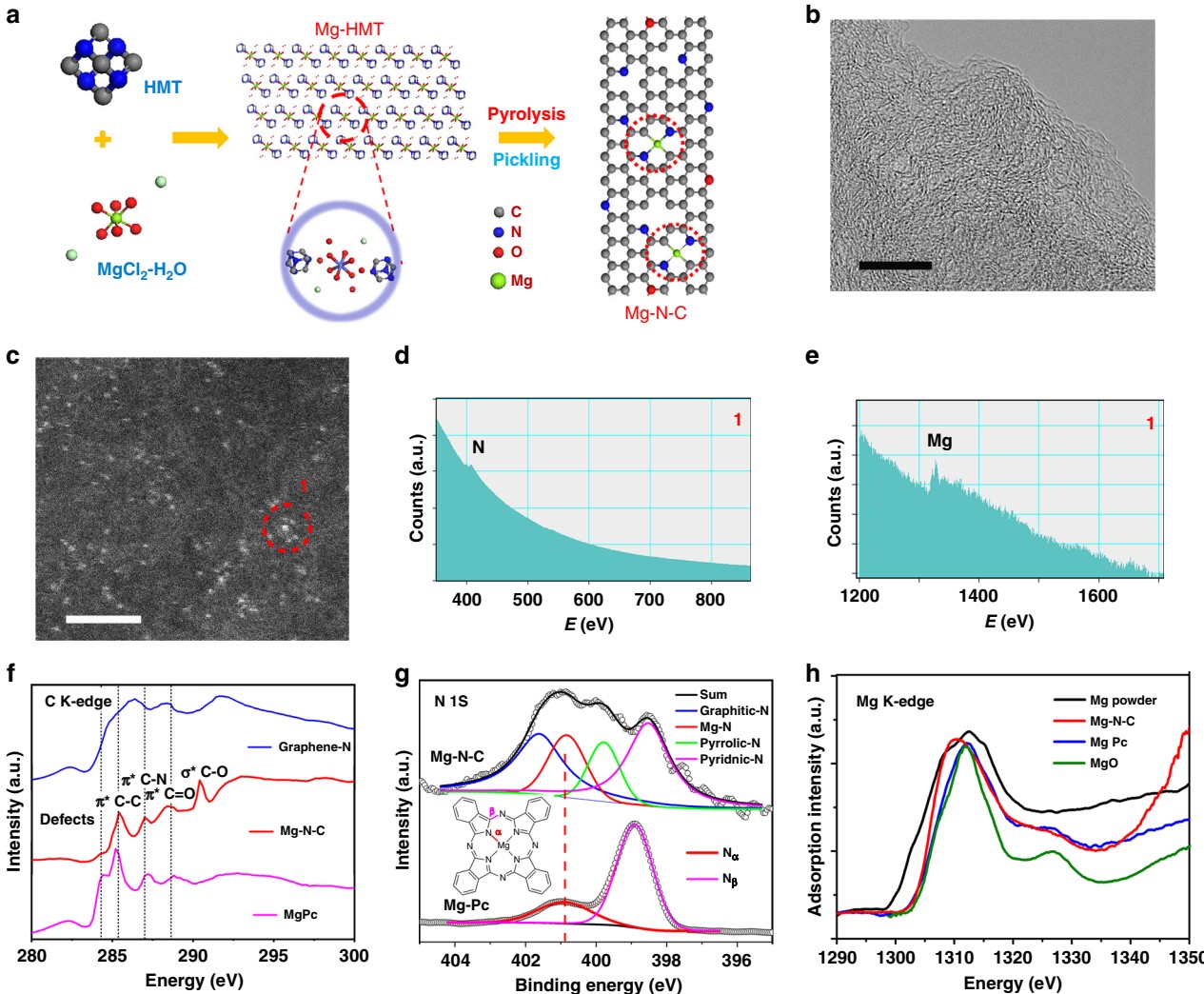

**Fig. 3 The synthesis and characterization of Mg–N–C. a** Schematic illustration of the synthesis procedure of Mg–N–C. The red dash circles at Mg–N–C represents the active centers. **b** High resolution transmission electron microscopy (HRTEM) of Mg–N–C. Scale bar: 5 nm. **c** High-angle annular dark-field-scanning transmission electron microscopy (HAADF-STEM) image of Mg–N–C. Scale bar: 2 nm. The bright dots are Mg atoms, which can be confirmed by electron energy loss spectroscopy (EELS) spectra in **d**, **e**, probe resolution is 0.15 nm. **d**, **e** The EELS spectrum for atomic site highlighted by red circle in **c**. **f** C K-edge XANES spectrum of Graphene-N, Mg–N–C and MgPc. **g** The XPS result of the N 1s spectrum for Mg–N–C and MgPc. **h** The XANES of Mg K-edge for samples.

High-angle annular dark-field atomic-resolution scanning transition electron microscopy (HAADF-STEM) images in Fig. 3c and Supplementary Fig. 21 reveal that there is large amount of atomic dispersed metal atoms (the bright dots) anchored in Mg–N–C. Electron energy loss spectroscopy (EELS) spectrums in Fig. 3d, e indicate that the bright dot in Fig. 3c at site 1 is corresponding to Mg–$N_x$ moieties. The Energy dispersive X-ray spectroscopy (EDS) images in Supplementary Fig. 17f shows that the elements of Mg, N, and C are uniformly distributed in Mg–N–C. The Mg content in Mg–N–C is up to 1.32wt% as determined by the inductively coupled plasma atomic emission spectroscopy (ICP-AES) analysis, and no extrinsic Fe, Co, Ni is detected (below 0.01wt%) (Supplementary Table 8).

The valence states of elements in Mg–N–C are investigated. X-ray absorption near-edge structure (XANES) in Fig. 3f reveals the catalyst is composed by heteroatoms modified carbon. The X-ray photoelectron spectroscopy (XPS) spectrum of N 1s in Fig. 3g can be deconvoluted into four peaks corresponding to pyridinic-N, pyrrolic-N, Mg–Nx (corresponding to $N_a$[38] of MgPc which bonds with Mg) and graphitic-N, respectively[30,39]. The Mg 1s and 2p

spectrum in Supplementary Fig. 22 reveals a +2 oxidation state of Mg and it can be deconvoluted into Mg–N and Mg–C bond, no Mg–O signal is detected. The coordination of Mg center in the catalyst is then further detected by XANES as can be seen in Fig. 3h. There is no obvious signal for metallic Mg at ~1304 eV in Mg–N–C sample, which agrees with the HAADF and EELS analysis results that most Mg are atomic bonded with nitrogen. The shape of absorption curve of Mg–N–C is distinct from those of Mg powder and MgO, while similar to that of Mg phthalocyanine (MgPc), which indicates that the valance state of Mg is between the metallic state (0) and oxidation state (+2). Note that most Mg-compounds including magnesium nitrides, oxides, and carbides (if they are not detected before pickling) will be removed after acid pickling. Thus only the macrocyclic structures, such as MgPc analogs composed by heat treatment remain in the final product[40,41]. Moreover, the edge spectrum of magnesium is sensitive to coordination number (CN), and the first resonance of K-edge at low energy which is related to the transitions from 1s to empty p states, will shift to lower energy as the CN decreases[42–44]. As shown in Fig. 3h, since the edge of

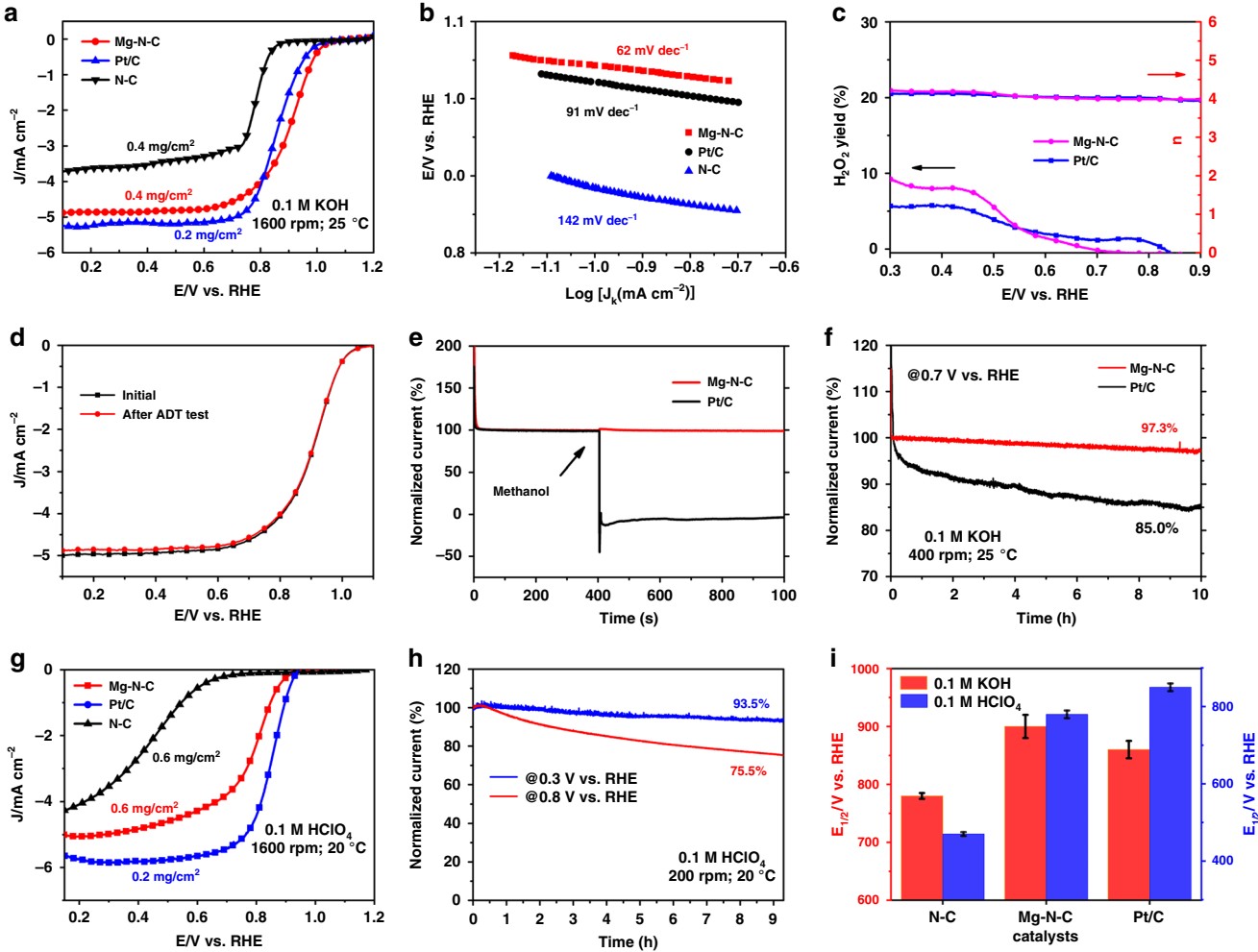

**Fig. 4 Performance of Mg–N–C as an oxygen reduction reaction catalyst. a** LSV curves after normalization by glass carbon (GC) electrode in O$_2$-saturated 0.1 M aqueous KOH electrolyte solutions at a sweep rate of 5 mV s$^{-1}$. **b** Tafel slope of the catalysts in 0.1 M KOH solution. **c** The H$_2$O$_2$ yield and the electrons transfer rate of the catalysts in 0.1 M KOH solution. **d** Mg–N–C before and after accelerated durability test (ADT) with 5000 cycles from 0.8 to 1.1 V (vs. RHE) at a sweep rate of 100 mV s$^{-1}$. **e** Methanol tolerance test of Mg–N–C at 0.8 V (vs. RHE) with 1600 rpm in O$_2$-saturated 0.1 M KOH solution. 4 ml methanol is injected into 110 ml O$_2$-saturated 0.1 M KOH solution at 400s. **f** CA test of the catalysts at 0.7 V (vs. RHE) with 400 rpm for 10 h in O$_2$-saturated 0.1 M KOH solution. **g** LSV curves after normalization by glass carbon (GC) electrode in O$_2$-saturated 0.1 M aqueous HClO$_4$ electrolyte solution at a sweep rate of 10 mV s$^{-1}$. **h** CA test of Mg–N–C in 0.1 M HClO$_4$ solution. **i** $E_{1/2}$ for different catalysts in both alkaline and acidic solutions.

Mg–N–C is left shifted comparing to that of MgPc (CN = 4), the number of CN should be less than 4 in Mg–N–C. Furthermore, according to DFT simulations, MN1C is an unstable cofactor and the activity of MN3C is too poor to catalyze ORR. Thus, the results of DFT and XANES for Mg–N–C indicate that the most likely high active moiety in Mg–N–C is the MN2C cofactor (the different spatial configurations similar to MN2C like isomer and edge-bonded moieties are compared in Supplementary Figs. 39–41).

The ORR activity of Mg–N–C is then investigated in both basic and acidic environments with a rotating disk electrode (RDE). For comparison, the ORR performance of nitrogen-doped graphene (N–C) and commercial 20 wt% Pt/C are also tested. Figure 4a shows the linear sweep voltammetry (LSV) curve after normalization by glass carbon (GC) electrodes in O$_2$-saturated 0.1 M aqueous KOH solutions. The Mg–N–C shows excellent ORR activity with half-wave potential ($E_{1/2}$) of 910 mV and onset potential ($E_{onset}$) of 1.03 V. The $E_{1/2}$ of Mg–N–C surpasses the value of 780 mV for N–C and is higher than that of 860 mV for commercial Pt/C. The outstanding ORR activity is further confirmed by the smaller Tafel slope of Mg–N–C with a value

of 62 mV dec$^{-1}$ as compared to that of 91 mV dec$^{-1}$ for Pt/C and 142 mV dec$^{-1}$ for N–C as shown in Fig. 4b. It indicates that the first electron transfer (59 mV dec$^{-1}$) catalyzed by Mg–N–C is probably the rate-determining step. The electrochemical impedance spectroscopy (EIS) diagram in Supplementary Fig. 23 shows smaller impedance of Mg–N–C as comparing to Pt/C. According to the rotating ring disk electrode (RRDE) test result shown in Fig. 4c, the electron-transfer rate for Mg–N–C is close to 4, similar to that of Pt/C. The H$_2$O$_2$ yield is approximated to be zero at 0.8 V, which proves the existence of a 4e$^-$ associative pathway for Mg–N–C in ORR (it's also confirmed by RDE voltammograms at various rotation rates in Supplementary Fig. 24). Accelerated durability test (ADT) is cycled in O$_2$-saturated 0.1 M KOH to test the durability. As shown in Fig. 4d, after 5000 cycles, negligible change in $E_{1/2}$ is observed, indicating the high stability of catalyst. According to chronoamperometry (CA) measurements in Fig. 4e, after injecting methanol into the solution, significant drop in the current density for Pt/C is observed, while Mg–N–C shows excellent tolerance to methanol crossover. Moreover, 97.3% normalized current is retained at 0.7 V (RHE) after 10 h durability test for Mg–N–C as can be seen

in Fig. 4f, while only 85.0% normalized current is retained for Pt/C, which also indicates great stability of the Mg–N–C catalyst.

The ORR activity of Mg–N–C in acidic condition is also evaluated. In contrast to the poor activity of most metal-free nitrogen-doped carbon-based catalysts in acidic condition[45,46], Mg–N–C shows excellent performance with $E_{1/2}$ up to 790 mV in 0.1 M HClO₄ as shown in Fig. 4g. High ORR stability is also observed in Fig. 4h. The activity of Mg–N–C with different loading is shown in Supplementary Fig. 25. Figure 4i summarizes the ORR performance of all three catalysts, indicating that Mg–N–C is close to that of Pt/C in acidic media while surpasses that of Pt/C in alkaline condition. A comparison of the ORR activity summarized in Supplementary Table 9 between Mg–N–C and other transition metal based catalysts reported so far shows that it even exceeds the ORR activity of most non Fe-based catalysts. Furthermore, when Mg–N–C is used to build zinc–air batteries, it shows dramatic stability under steady discharge at current density of 20 mA cm$^{-2}$ for 8 h as can be seen in Supplementary Fig. 26. Ca-HMT and Al-HMT are also synthesized and tested. Consistent with the DFT simulations, the ORR activity of these pyrolysis products are much less active than Mg–N–C as shown in Supplementary Fig. 27.

According to DFT simulations and experimental results, we conclude that Mg–N–C shows outstanding ORR performance which is comparable to Pt/C and some transition metals. In order to find out the active site in Mg–N–C corresponding to the catalytic activity, in Supplementary Fig. 29, we explore its activity contribution from high electrochemical surface area (ECSA) and evaluate its activity by high kinetic current density ($J_k$). Supplementary Fig. 30 shows that impurity of amorphous oxide before acid pickling is not the active mass since the ORR activity is enhanced after etching treatment. A poisoning experiment is then performed by employing KSCN to block the M-Nx sites[47–49]. As shown in Supplementary Fig. 31, the $E_{1/2}$ with negative shifts by 109 mV due to the poisoning effect indicates that Mg–N moiety is the active site in ORR.

The Mg content in Mg–N–C is about 1.3wt% and the ratio of Mg–N is hard to be varied due to the constant Mg ratio in metal-ligand in MOFs precursor. To test the influence of Mg doping content on catalytic activity, another experiment is designed to vary Mg content in nitrogen doped carbon (N–C) host[50]. The detailed synthesis process is described in Supplementary Information. According to Supplementary Fig. 32, after heat treatment, the ORR activity in both acidic and basic electrolytes are improved with increasing Mg (+2) contents. Structure characterization in Supplementary Figs. 33–36, confirms that the increasing content of Mg (+2) cooperated with N-doped carbon matrix, but neither the N-doping nor modified carbon, is responsible for activity enhancement. These results confirm that it is the Mg–N moiety corresponding to the high ORR activity of Mg–N–C.

We then further explore the possible reaction pathway of MN2C in ORR. The initial adsorbed sites are set at Mg, N, C1 adjacent N and C2 away from N (labeled MN2C-M, MN2C-N, MN2C-C1, and MN2C-C2, respectively). Supplementary Fig. 37 shows that adsorbates would all move to the Mg sites for the final stable structures, which confirms the Mg sites are the preferred stable adsorbate sites of these models. The free energy diagram of different models as can be seen in Fig. 5a indicates that the most energetic optimized catalytic site is MN2C-C1, where the rate determine step (RDS) is OOH* formation step with the lowest energy barrier value of 0.41 eV in case of 1.23 V output potential (thus leads to the highest value of $U_{RHE}^{onset}$ = 1.23 V − 0.41 V = 0.82 V). The reaction pathway of MN2C-C1 are shown in Fig. 5b i–v. Notable, O* is adsorbed to both Mg and C1 as shown in Fig. 5b iv, which reveals

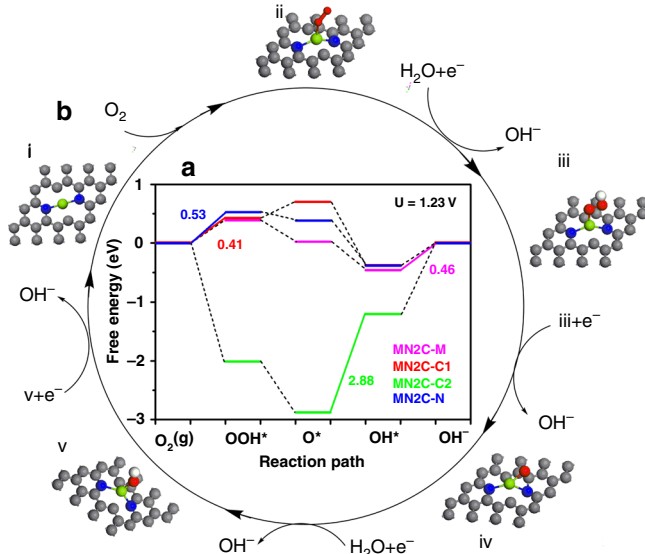

**Fig. 5 The free energy diagram of MN2C and its possible reaction pathway. a** The free energy diagram of MN2C at different optimized adsorbed reaction sites at 1.23 V and the full oblique lines and corresponding value shows the RDS and energy barriers of catalysts. **b** Favorable reaction pathway of MN2C. And the active sites are Mg and C1 atoms. Green, blue, red, white and gray balls represent metal, N, O, H, and C atoms, respectively. Specially, the O* is adsorbed at both Mg and C1 adjacent to N atoms in (iv).

that the dual catalytic sites in MN2C moiety (see more in Supplementary Figs. 37–38). The lowest reaction energy barrier of MN2C in ORR than that of other main group metal cofactors while closed to that of Pt, makes it reasonable excellent performance.

## Discussion

In summary, we report, for the first time, a high-performance Mg-based electrocatalyst with a super high ORR activity surpassing that of the commercial Pt/C in alkaline electrolyte and being comparable to that in acidic condition. It changes our common knowledge of catalytic properties of main group metals-based materials. DFT calculations reveal that a higher p-state location is generated as a Mg center is coordinated with two nitrogen atoms in the graphene matrix compared to other coordination numbers, which weakens the oxygenated species binding strength at Mg atom and leads to an ORR activity near the top of volcano-type activity plots. This study indicates that the rational materials design can help to fabricate highly active electrocatalysts based on main group metals, which may shed light on further development of catalysts.

## Methods

**Pre-doped graphene-based N-coordinated Mg cofactors.** Mg-based MOFs:[Mg (H₂O)₆]Cl₂·2[(CH₂)₆N₄]·4H₂O is prepared as follows. The samples are synthesized in water-alcohol mixed solution (ratio = 1:2) at room temperature (about 28 °C). Briefly, the 7.14 mM MgCl₂·6H₂O and 3.57 mM hexamethylenetetramine are dissolved into the mixed solution (10 ml alcohol and 5 ml water) separately and then the solutions are mixed together. After magnetic stirring the liquid at room temperature for half an hour, snowflake-like solid is precipitated. The precipitate is dried in the oven at 80 °C for overnight and the final product is obtained, denoted as Mg-HMT.

Mg–N–C is prepared as follows. Mg-HMT is pyrolyzed in heat treatment at 900 °C under Ar atmosphere for 1 h with heating rate of 5 °C min$^{-1}$. The pyrolysis product is gray fragile bulky grains, ground by the grinding bowl. In order to get rid of the impurities, the product is then pickled in 3 M HCl aqueous solution and washed with water and ethanol (the effect from acid treatment is discussed in section 2.2 in Supplementary Information). After drying, black Mg–N–C powder is obtained. It should be noted that appropriate drying for the precursor is crucial

before and during the pyrolysis process. The product will turn into tiny black particles after being ground, otherwise, graphite (002) strong peak will exist in the XRD pattern of Mg–N–C products if the material is not fully ground as can be seen in Supplementary Fig. 18.

The synthesis process of Al-HMT and Ca-HMT are similar to that of the Mg-HMT except that AlCl₃·6H₂O and Ca(NO₃)₂·6H₂O are used.

For comparison, graphene-N is synthesized using a modified Hummers' method[51] and annealed under NH₃ atmosphere.

**Post-doped graphene-based N-coordinated Mg cofactors**. ZIF-8 based N–C-pristine is prepared as follows. 1.68 g hydrated Zn(NO₃)₂ and 4 g 2-methylimidazole are dissolved into 20 ml and 60 ml methanol at room temperature. The two solutions are then mixed together and stirred for 1 h. A centrifugal precipitation is obtained, denoted as ZIF-8. After pyrolysis at 900 °C with heating rate of 5 °C min⁻¹ for 3 h under N₂ atmosphere and then pickling in 3 M HCL, ZIF-8 is transferred into nitrogen-doped porous carbon (**N–C pristine**).

Different contents of metal-N co-doped N–C are prepared as follows. 50 mg N–C-pristine, 30 mg MgCl₂·6H₂O and 100 mg melamine are dispersed into the isopropanol–water solution by a ratio of 1:1. After ultrasonic dispersion for 1 h, the mixture is stirred for 3 h and a centrifugation is obtained. The centrifugal product is dried at 60 °C in the oven for at least 12 h and pyrolyzed at 900 °C with heating rate of 5 °C min⁻¹ for 2 h in Ar atmosphere, leading to the formation of isolated single Mg atom anchored on nitrogen species (**N–C–1Mg**). **N–C–0Mg** (also called **N–C–no metal**), **N–C–3Mg** and **N–C–3Al** are also obtained following the same procedure except that 0 mg MgCl₂·6H₂O, 100 mg MgCl₂·6H₂O and 120 mg AlCl₃·6H₂O (the same mole ratio with Mg) are used. **N–C–3Mg-no N source** is also synthesized by the same procedure as N–C–3Mg except that no N source (melamine) is added.

**Material characterization**. The HRTEM (JEOL-2011) is operated at an accelerating voltage of 200 kV. The high-angle annular dark-field scanning transmission electron microscopy (HAADF-STEM) and corresponding energy-dispersive spectroscopy (EDS) mapping analyses are executed on JEOL JEM-ARF200F TEM/STEM with a spherical aberration corrector. TGA is carried out using Shimadzu-50 thermoanalyser under flowing Ar atmosphere with a heating rate of 10 °C min⁻¹. The XPS is performed on ESCALAB 250 X-ray photo-electron spectrometer using Al Kα radiation. The X-ray absorption near edge structure (XANES) of Mg K-edge is investigated at the BL08U1A beamline of Shanghai Synchrotron Radiation Facility (SSRF). The adsorption curves are treated by normalized process.

**Electrochemical measurements**. Due to the large scale of the catalyst, full grinding is treated before test. ORR measurements are carried out using conventional rotating disk method with a MSR Electrode Rotator (Pine Research Instrumentation). A glassy carbon (GC) electrode (diameter of 5 mm with surface area of 0.196 cm²) is used as a working electrode. Au plate (1cm*1cm) is utilized as the counter electrode in alkaline condition, instead of Pt to prevent pollution from longtime electrochemical corrosion, and graphite rod is used in acidic media. There is no change for initial ORR performance from different counter electrodes used in test is shown in Supplementary Fig. S28. The potential is recorded using Ag/AgCl (3.5 M KCl) electrode as the reference electrode. All of the potentials are converted to the reversible hydrogen electrode (RHE) according to equation: E(vs. RHE) = E (vs. Ag/AgCl) + 0.198 V + 0.059 × pH.

In 0.1 M KOH solution, the loading amount of Mg–N–C and N–C is 0.4 mg/cm². Four milligrams of catalyst and 30 μL Nafion solution (Sigma Aldrich, 5 wt %) were dispersed in 1 mL ethanol solution. Electrochemical measurements for the Mg–N–C catalyst are first conducted in N₂-saturated 0.1 M KOH solution and then under O₂-saturated solution. The ORR curve is acquired under oxygen-saturated conditions by subtracting the data under nitrogen-saturated conditions as a background. Electrochemical measurements are conducted from 1.2 to 0 V (vs. RHE) with a scanning rate of 0.005 V/s. The CV test is performed with a scanning rate of 0.01 V/s. The commercial Pt/C is used as the reference material with the loading of 0.2 mg/cm² with a scanning rate of 0.01 V/s. The electron transferred number ($n$) is calculated according to the Koutecty–Levich (K–L) plots linear fit lines from the K–L equation:

$$\frac{1}{J} = \frac{1}{J_K} + \frac{1}{J_L} = \frac{1}{J_K} + \frac{1}{Bw^{\frac{1}{2}}}$$

$$B = 0.62nFC_0D_0^{2/3}V_0^{-1/6}$$

where $J$ is the measured current density, $J_K$ and $J_L$ are the kinetic- and diffusion limiting current density, $\omega$ is the angular velocity, n is transferred electron number, F is the Faraday constant (96485 C mol⁻¹), $D_0$ is the diffusion coefficient of O₂ (1.9 × 10⁻⁵ cm² s⁻¹), $C_0$ is the bulk concentration of O₂ (1.2 × 10⁻⁶ mol cm⁻³) and V is the kinematic viscosity of the electrolyte (0.01 cm² s⁻¹).

The hydrogen peroxide yield (H₂O₂ %) and the electron-transfer number (n) are calculated using the following equations:

$$H_2O_2(\%) = 200 \times \frac{\frac{I_R}{N}}{I_D + \frac{I_R}{N}}$$

$$n = 4 \times \frac{I_D}{\frac{I_R}{N} + I_D}$$

where $I_D$ is the disk current, $I_R$ is the ring current, and N is the ring collection efficiency with a value of 0.4.

In 0.1 M HClO₄, the loading amount of Mg–N–C and N–C catalysts is 0.6 mg/cm² and is 0.2 mg/cm² for Pt/C. Electrochemical measurements are conducted from 1.2 to 0 V (vs. RHE) with a scanning rate of 0.01 V/s. Poison test is carried out by adding KBr, KCl, K₂SO₄, and KSCN into the solution with same mole quantities (10 mM).

The ESCA value is calculated by the equation: $ECSA = \frac{C_{dl}}{C_s}$

Where $C_{dl}$ is the value of electrochemical double layer capacitance. The ECSA is measured on the same working electrode and electrolyte (0.1 M KOH). The potential window of CVs was 1–1.1 V vs. RHE, and the scan rates are 5, 10, 15, 20, and 25 mV/s. The $C_{dl}$ is estimated by plotting the Δj at 1.05 V vs. RHE against the scan rate. The slope is twice of $C_{dl}$. Note that the ECSA calculated from the $H_{upd}$ is suitable for the precious metal catalysts but is not reasonable for carbon-based catalysts for the corresponding ECSA usually include the area of carbon which is not the intrinsic active sites[52,53]. So $C_{dl}$ is used to represent the ECSA and is shown in Supplementary Fig. S29a.

Due to the ECSA value from this method is inaccurate (always much higher than real reaction active sites) caused from extra carbon active sites, the standardized current density from ECSA $\left( j_{ECSA} = \frac{j}{A_{ECSA}} \right)$ is underestimated, so we only the standardized current density from geometric area $\left( j_{Geom} = \frac{j}{A_{Geom}} \right)$ to evaluate the performance of catalysts.

**Zinc–air battery tests**. A home-made zinc–air battery device is designed for the performance and stability test. 6 M KOH is the electrolyte while the polished zinc plate is used as the anode. Mg–N–C catalyst ink is brushed onto a 1 cm² carbon onto 1 cm² carbon paper (HCP030) as cathode. 20 wt% Pt/C and Graphene-N (N–C) catalysts are also prepared following the same procedure.

The electrolyte used in the primary zinc–air battery is 6 M KOH. A polished zinc plate was used as the anode and a certain volume of Mg–N–C catalyst ink is brushed onto 1 cm² carbon paper (HCP030) as cathode with a catalyst loading of 2 mg cm⁻². As reference materials, 20 wt% Pt/C and Graphene-N (N–C) catalysts were also prepared with the same procedure and loading amount.

**DFT calculation**. All the DFT calculations are performed using the Vienna Ab Initio Simulation Package (VASP). The exchange-correlation potential is described by the generalized gradient approximation (GGA) with spin polarized Perdew–Burke–Ernzerhof (PBE) functional. The projector augmented wave is applied to describe the electron-ion interaction and the plane-wave energy cutoff is set to 400 eV. All structures are optimized with a convergence criterion of $1 \times 10^{-5}$ eV for the energy and 0.01 eV/Å for the forces. A periodic $4 \times 4$ graphene support is built. The vacuum spacing is set to more than 15 Å for surface isolation to prevent interaction between two neighboring surfaces. Brillouin zone sampling is employed using a Monkhorst-Packing grid with $9 \times 9 \times 1$ for the calculated models. Denser k-points ($9 \times 9 \times 1$) are used for the calculations of density of states (DOS). For commercial Pt/C, we use a Pt ($3 \times 3$) unit cell of Pt (111) surface models. The Pt (111) slab has 4 atom layers and the top two layers are fully relaxed during the structural optimization and geometry optimizations for Pt (111) are performed with $4 \times 4 \times 1$ k-mesh.

The ORR pathways on metal cofactors systems were calculated in detail according to the electrochemical framework developed by Nørskov and his co-workers[54]. For ORR, in an alkaline electrolyte, H₂O rather than H₃O⁺ may act as the proton donor, so overall reaction scheme of the ORR can be written as:

$$O_2 + 2H_2O + 4e^- \leftrightarrow 4OH^-$$

The ORR may proceed through the following elementary steps which are usually employed to investigate the electrocatalysis of the ORR on various materials:

Therefore, we took reactions (1)−(4) to derive the thermochemistry for ORR.

$$O_2(g) + H_2O(l) + e^- + * \rightarrow OOH^* + OH^- \tag{1}$$

$$OOH^* + e^- \rightarrow O^* + OH^- \tag{2}$$

$$O^* + H_2O(l) + e^- \rightarrow OH^* + OH^- \tag{3}$$

$$OH^* + e^- \rightarrow * + OH^- \tag{4}$$

where * stands for an active site on the catalytic surface, (l) and (g) refer to liquid and gas phases, respectively.

The reversible hydrogen electrode (RHE) model developed by Nørskov and co-workers was used to obtain Gibbs reaction free energy of these electrochemical elementary steps. In this model, we set up RHE as the reference electrode, which allows us to replace chemical potential ($\mu$) of proton–electron pair with that of half a hydrogen molecule:

$$\mu_{H^+} + \mu_{e^-} = \frac{1}{2}\mu_{H_2}$$

at conditions with U = 0 V and $P_{H_2}$ = 1 bar.

The chemical potential of each adsorbate is defined as:

$$\mu = E + E_{ZPE} - TS$$

where the $E$ is the total energy obtained from DFT calculations, $E_{ZPE}$ is zero-point energy and $S$ is the entropy at 298 K.

Since it is difficult to obtain the exact free energy of OOH, O, OH radicals in the electrolyte solution, the adsorption free energy $\Delta G_{OOH^*}$, $\Delta G_{O^*}$ and $\Delta G_{OH^*}$ are relative to the free energy of stoichiometrically appropriate amounts of $H_2O$ (g) and $H_2$ (g), defined as follows:

$$\Delta G_{OOH^*} = \Delta G\left(2H_2O(g) + * \rightarrow OOH^* + \frac{3}{2}H_2(g)\right)$$
$$= \left(E_{OOH^*} + 1.5 \times E_{H_2} - 2 \times E_{H_2O} - E^*\right)$$
$$+ \left(E_{ZPE}(OOH^*) + 1.5 \times E_{ZPE}(H_2) - 2 \times E_{ZPE}(H_2O) - E_{ZPE}(*)\right)$$
$$- T \times \left(S_{OOH^*} + 1.5 \times S_{H_2} - 2 \times S_{H_2O} - S_*\right)$$

$$\Delta G_{O^*} = \Delta G(H_2O(g) + * \rightarrow O^* + H_2(g))$$
$$= \left(E_{O^*} + E_{H_2} - E_{H_2O} - E^*\right)$$
$$+ \left(E_{ZPE}(O^*) + E_{ZPE}(H_2) - E_{ZPE}(H_2O) - E_{ZPE}(*)\right) - T$$
$$\times \left(S_{OOH^*} + S_{H_2} - S_{H_2O} - S_*\right)$$

$$\Delta G_{OH^*} = \Delta G\left(H_2O(g) + * \rightarrow OH^* + \frac{1}{2}H_2(g)\right)$$
$$= \left(E_{OH^*} + 0.5 \times E_{H_2} - E_{H_2O} - E^*\right)$$
$$+ \left(E_{ZPE}(OOH*) + 0.5 \times E_{ZPE}(H_2) - E_{ZPE}(H_2O) - E_{ZPE}(*)\right) - T$$
$$\times \left(S_{OOH^*} + 0.5 \times S_{H_2} - S_{H_2O} - S_*\right)$$

For each elementary step, the Gibbs reaction free energy $\Delta G$ is defined as the difference between free energies of the initial and final states and is given by the expression:

$$\Delta G = \Delta E + \Delta ZPE - T\Delta S + \Delta G_U + \Delta G_{PH}$$

where $\Delta E$ is the reaction energy of reactant and product molecules adsorbed on catalyst surface, obtained from DFT calculations; $\Delta ZPE$ and $\Delta S$ are the change in zero point energies and entropy due to the reaction. $\Delta G_U = -neU$, where $U$ is the electrode applied potential relative to RHE as mentioned above, $e$ is the elementary charge transferred and $n$ is the number of proton–electron pairs transferred. $\Delta G_{PH}$ is the correction of the $H^+$ free energy $\Delta G_{PH} = -k_B Tln[H^+] = pH \times k_B Tln10$. Hence, the equilibrium potential $U_0$ for four-electron transfer ORR at pH = 14 was determined to be 0.402 V vs NHE and 1.23 V vs RHE according to Nernst equation: $E = E_0 - 0.0591$ pH, $U_0(RHE) = U_0(NHE) + 0.828$ V = 0.402 + 0.828 = 1.23 V), where the reactant and product are at the same energy level. Due to the difficulty to describe the oxygen molecule with high-spin ground state, the free energy of the $O_2$ molecule was derived according to:

$$G_{O^2}(g) = 2G_{H_2O}(l) - 2G_{H_2} + 4 \times 1.23(eV)$$

The reaction free energy of (1)–(4) for ORR can be calculated using the following equations:

$$\Delta G_1 = \Delta G_{OOH^*} - 4.92$$

$$\Delta G_2 = \Delta G_{O^*} - \Delta G_{OOH^*}$$

$$\Delta G_3 = \Delta G_{OH^*} - \Delta G_{O^*}$$

$$\Delta G_4 = -\Delta G_{OH^*}$$

The onset potential is calculated by:

$$U_{RHE}^{onset} = -max\{\Delta G_1, \Delta G_2, \Delta G_3, \Delta G_4\}$$

In this paper, we focus on the adsorbates strength on alkaline condition. The free energy of elementary step in acidic condition can be transform by the Nernst equation, but it doesn't affect the trend of adsorbates strength on metal sites.

The over potential $\varnothing$ (V vs. RHE) in ORR test is calculated by:

$$\varnothing = 1.23 - U_{RHE}^{onset}.$$

The higher value of $U_{RHE}^{onset}$ is, the lower value of $\varnothing$ is, and thus better predicted ORR activity is.

The volcano map and volcano plot: They are visible predictions about the trend of reaction activity. In this manuscript, based on the difference of adsorption strength at kinds of metal sites, we calculated the relationship between intermediates adsorption ($\Delta G_{OOH}$, $\Delta G_{O^*}$ and $\Delta G_{OH^*}$), linear relationship can be fitted by:

$$\Delta G_{OOH^*} = 1.008 \times \Delta G_{OH^*} + 3.444 (R^2 = 0.997)$$

$$\Delta G_{O^*} = 0.94 \times \Delta G_{OH^*} + 2.606 (R^2 = 0.977)$$

The onset potentials for each model is correlated with the free energy of intermediates adsorption, which can be calculated by:

For ORR, the onset potential is calculated by:

$$U_{RHE}^{onset} = -max\{\Delta G_1, \Delta G_2, \Delta G_3, \Delta G_4\}$$

where $\Delta G_1$, $\Delta G_2$, $\Delta G_3$, $\Delta G_4$ are the free energies of reaction barriers of each elementary steps, and calculated by:

$$\Delta G_1 = \Delta G_{OOH^*} - 4.92$$

$$\Delta G_2 = \Delta G_{O^*} - \Delta G_{OOH^*}$$

$$\Delta G_3 = \Delta G_{OH^*} - \Delta G_{O^*}$$

$$\Delta G_4 = -\Delta G_{OH^*}$$

Therefore, the free energy of elementary steps and theoretical activity can be transformed to be the $\Delta G_{OH^*}$-related quantities.

With these equations, volcano plot can be drawn.

For volcano map, the free energy of elementary steps and theoretical activity can be transformed to be the variables related to $\Delta G_{OOH^*}$ and $\Delta G_{OH^*}$, and a three-dimensional volcano map is drawn. And the date dots represented the main group metal cofactors are located at the projection of volcano map.

The definition of p-band center of metal atoms in cofactors: For the calculation of the band center of metal atoms projected on sum of the p-orbital of metal atom, the following expression is used:

$$\varepsilon_P = \frac{\int_{-\infty}^{+\infty} E \times \rho_P dE}{\int_{-\infty}^{+\infty} \rho_P dE}$$

where $\rho_P$ is the density of p-state projected onto metal-atom.

## Data availability

The data that support the findings of this study are available from the corresponding author upon reasonable request.

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

## Acknowledgements

This study was supported by the National Natural Science Foundation (51772283), the National Key R&D Program of China (Grant No. 2016YFA0401801) and Fundamental Research Funds for the Central Universities (WK2060140021). The calculations were completed on the supercomputing system in the Supercomputing Center of USTC. We thank BL08U1A beam station for XANES test in Shanghai Synchrotron Radiation Facility (SSRF). We thank the soft X-ray magnetic circular dichroism end station (XMCD) of the National Synchrotron Radiation Laboratory (NSRL) for measuring C K-edge and N K-edge XANES.

## Author contributions

Shuai Liu and Qianwang Chen conceived the idea, designed research and analysed the data. Shuai Liu, Zedong Li, and Changlai Wang performed the DFT calculations; Shuai Liu and Minxue Huang synthesized the catalysts and conducted the reaction tests; Ming Zuo conducted the STEM and EELS analyses; Lijuan Zhang performed the XANES analysis. Yang Yang, Yang Kang, Shi Chen, and Pengping Xu performed the structural characterization analysis. Shuai Liu, Weiwei Tao and Qianwang Chen wrote the paper.

## Competing interests

The authors declare no competing interests.

## Additional information

**Peer review information** *Nature Communications* thanks the anonymous reviewers for their contributions to the peer reivew of this work. Peer review reports are available.

