## [Peer Review File · Nature Communications]

Reviewers' comments:

Reviewer #1 (Remarks to the Author):

In this manuscript, the authors have found a newly strategy to improve the ORR performance of main group metals by changing the coordination environment. Both calculation and experimental results verified that the MgN₂C catalyst has an outstanding ORR activity, which is comparable to Pt/C in both alkaline and acidic solutions. The paper is interesting and publishable after considering the following questions.

1. The DFT simulations in this work are mainly focused on M-N-C local structure catalysts. As we know, the real coordination environment of the active site could be much more complex and may impact its catalytic performance, eg. pyrrolic N, pyridinic N, graphitic N, hole, etc. The experimental results of XPS in Fig. 3g shown that it was more likely to be the pyrrolic N, the authors should give more explanations. At least, authors needs to confirm the existence of metal-carbon bonds
2. The p-band center was used to explain the key intermediate adsorption strength on metals sites. Can the authors give a definition of the p-band center and its calculation method? Can the p-center position be used as a descriptor to describe electrocatalytic activity?
3. The theoretically calculated onset potential is anything related to the measured value in the experiment? Can the adsorption energy of *OH (ΔG_{OH^*}) also works well in describe the experimental results?
4. Personally, I am quite confused by Fig. 4i and Table S9, the E_{1/2} for Mg-N-C and Pt/C in both alkaline and acidic solutions. In table S9, the E_{1/2} is 910 mV and 790 mV for Mg-N-C in alkaline and acidic solutions, and both of the values are higher than that for Pt/C (860 mV in alkaline of 780 mV in acidic). Otherwise, the ORR performance of Mg-N-C and Pt/C in alkaline is higher than that in acidic. medium However, there is an opposite conclusion in Fig. 4i?
5. The calculation results used to establish a volcano map are too less in Fig 2d. For an ideal volcano should include both side location and summit of volcano map.
6. The result of EIS in Pt/C was missing in Fig.S23.
7. The Fig. 6 should be Fig. 4.
8. There are already some reports on catalytic activity of main metal group N-C catalysts that should be included (e.g., *Advanced Materials*, 29, 1606635 (2017))
9. The author calculated the onset potential for the catalysts and compared with Pt. how did the authors include the effect of electrolytes (alkaline and acidic).
10. In nature, Chlorophyll contains MgN₄ with Mg²⁺..but in this report, M-N₂C is considered as the most active center with Mg²⁺. There may be mixed bond configuration in MgN₂C.

Reviewer #2 (Remarks to the Author):

The article entitled "Learning from Nature, a case to turn main group element Mg into a highly active electrocatalyst for oxygen reduction reaction" deals with DFT calculation and experimental study of M-N-C type of materials for ORR.

Despite the high interest to the subject in the field this article does not contribute to the field on the exceptional level required by Springer Nature Publishing group. Material has extremely low performance in acidic media. Performance in alkaline is similar to previously published data. Synthetic approach was presented decades ago.

Decision – Decline.

Reviewer #3 (Remarks to the Author):

In this manuscript, the catalytic activity of Mg-based electrocatalyst for the ORR is reported in acidic and basic electrolytes. The Mg-based electrocatalyst offers high ORR activity surpassing that of the Pt/C commercial electrocatalyst.

- 1) Please revise the XPS plots shown in Fig. 3 (g) for the Mg-N-C for improving the position of the fit with the experimental data points.
- 2) Please make sure that all figures are correctly numbered and labeled. Please replace figure number 6 with figure 4.
- 3) Line number 264: Here, Fig. 5B is mentioned, while this figure number is not seen in the manuscript.
- 4) The reference number 10 and 16 are similar. However, their numbers are different.
- 5) The reference style is not according to the journal requirement, and it is not consistent.
- 6) Why MN3C and MN4C exhibits lower ORR activity compared to MN2C?
- 7) Please describe why the Mg sites are active compared to Al even though both transforms O₂ to OOH radical are exothermic.

Response to the comments

Dear Editor and Reviewers:

Thanks for your letter and for the reviewers' comments concerning our manuscript entitled “**Learning from Nature, a case to turn main group element Mg into a highly active electrocatalyst for oxygen reduction reaction**” (Manuscript ID: NCOMMS-19-22013-T). Those valuable and professional comments are very useful for us to revise and improve our manuscript. We have studied all the comments carefully.

The corrections are highlighted with blue color in the revised main article and supplementary information.

[Reviewer 1]

In this manuscript, the authors have found a newly strategy to improve the ORR performance of main group metals by changing the coordination environment. Both calculation and experimental results verified that the MgN₂C catalyst has an outstanding ORR activity, which is comparable to Pt/C in both alkaline and acidic solutions. The paper is interesting and publishable after considering the following questions.

Comments 1:

The DFT simulations in this work are mainly focused on M-N-C local structure catalysts. As we know, the real coordination environment of the active site could be much more complex and may impact its catalytic performance, eg. pyrrolic N, pyridinic N, graphitic N, hole, etc. The experimental results of XPS in Fig. 3g shown that it was more likely to be the pyrrolic N, the authors should give more explanations. At least, authors needs to confirm the existence of metal-carbon bonds.

Reply 1:

Thank you for this suggestion. In **Fig. 3g**, as the reference, the N 1s spectrum of MgPc can be deconvoluted into N_{α} (corresponding to Mg-N-C bond) and N_{β} (corresponding to C-N-C bond)¹. For our material Mg-N-C, the N 1s spectrum can be deconvoluted into four peaks corresponding to pyridinic-N, pyrrolic-N, Mg-N_x (corresponding to N_{α} of MgPc which bonds with Mg and C) and graphitic-N, respectively^{2,3}. Therefore, according to the XPS result of N 1s spectra we can only suggest the formation of N_{α} (Mg-N bond) in Mg-N-C catalyst. Indeed, we have tried to probe the coordination structure by X-ray absorption near-edge structure (XANES), but it is difficult to identify the specific coordination form (such as pyrrolic N or pyridinic N) from the synchrotron radiation technology for these light elements atoms, such as C, N and Mg.

Fig. 3g. The XPS result of the N 1s spectrum for Mg-N-C and MgPc.

While the XPS results of Mg 2p and 1s spectrum give more information. As shown below in **Figure S22a**, the Mg 1s spectra can be deconvoluted into two peaks, one is Mg-N bond (at 1304.2eV referred to MgPc), another peak located at 1305.2eV matches well with the literature reported Mg-C bond⁴. And also there is clearly two peaks in the Mg 2p spectra which can be deconvoluted into Mg-N bond (at 50.6eV referred to MgPc) and Mg-C bond (at 52.6eV^{5,6}). Note that, as referred to NIST databases, there is no signal of Mg-O bond (the positions of 1303.9eV for Mg 1s and 50.25eV for Mg 2p) and no metallic Mg (the positions of 1303-1303.5eV for Mg 1s and 49.3-49.7eV for Mg 2p) in the XPS results as shown in **Figure S22**.

Fig. S22. The valence state of elements in Mg-N-C catalyst from XPS. (a and b) Mg 1s and 2p spectra in Mg-N-C and Mg-Pc as referred. (c) C 1s spectrum. (d) The O 1s signal from XPS before and after pickling.

Therefore, we can confirm the Mg atoms form the Mg-N and Mg-C bonds in the catalysts.

In one hand, from the EELS, XANES analysis and XPS results, we can confirm the Mg atom is coordinated by less than four N atoms on graphic matrix and bonds with N and C atoms (see manuscript 10 line 250). So we calculated the possible structure with different N and C coordinated Mg moieties and found MN₂C is theoretical active in ORR and it is compliance with the results of the experiments. Therefore, we think the MN₂C structure is the most likely high active moiety in the materials.

In the other hand, there may exist the other bond configurations (such as Mg atom coordinated by pyrrolic N), we can't distinguish by the existing tests and it needs to be further studied. However, it doesn't affect the conclusion that Mg-based catalyst can also be tuned to a highly active catalyst in ORR.

We had revised the manuscript and supplementary information to add the analysis of Mg-C bond in manuscript page 10 line 255 and supplementary information page 29 line 495.

In addition, we would like to calculate the catalytic activity of Mg cofactor

coordinated by pyrrolic N to compare with the activity of MN₂C cofactor if you can give us enough time to revise the manuscript.

Comments 2:

The p-band center was used to explain the key intermediate adsorption strength on metals sites. Can the authors give a definition of the p-band center and its calculation method? Can the p-center position be used as a descriptor to describe electrocatalytic activity?

Reply 2:

Yes, we have revised the calculation method and give the definition of the p-band center in supplementary information page 10 line 275 and it has been copied below.

We had tried to build a descriptor to describe the intermediate adsorption strength of OH* (ΔG_{OH^*}), and we found the trend that the higher p-band center position of Mg, Al or Ca is, the weak adsorption strength is. The difference is, for the Mg and Al cofactors it is obviously that the ideal ΔG_{OH^*} is located at the 6 eV of p-band center, while the ideal ΔG_{OH^*} is located at -4 eV for the Ca cofactors. The difference may cause by the influence of empty 3d orbital of Ca atom. So we only conclude the trend, but not the descriptor.

And more, there are some papers reported the descriptor of valence-band energy level to describe the graphene-H system in HER^{7,8}, but they also didn't give the quantified data about the descriptor, so we think there is still a long way to explore the main group catalysis.

The definition of p-band center of metal atoms in cofactors:

For the calculation of the band center of metal atoms projected on sum of the p-orbital of metal atom, the following expression is used:

$$\varepsilon_p = \frac{\int_{-\infty}^{+\infty} E \times \rho_p dE}{\int_{-\infty}^{+\infty} \rho_p dE}$$

where ρ_p is the density of p-state projected onto metal-atom.

Comments 3:

The theoretically calculated onset potential is anything related to the measured value in the experiment? Can the adsorption energy of *OH (ΔG_{OH^*}) also works well in describe the experimental results?

Reply 3:

DFT calculation is a powerful tool to reveal the potential active catalysts and give the trend of catalytic activity of models.

In this study, we calculated the onset potential to evaluate the potential catalytic activity. From the viewpoint of thermodynamics, the catalytic activity (onset potential)

for ORR is strongly related to adsorption strength of oxygen-bearing intermediates (O^* , OH^* , and OOH^*) at catalytic sites^{9,10}, so the onset potential can be calculated by DFT with the free energy of intermediates adsorption on catalyst. The catalytic activity calculated in this work indicates that the onset potential (0.78V) of MgN2C (denoted as MN2C) is higher than that of Co-N4-C (0.73V) but lower than Fe-N4-C (0.81V). However, we must note that the DFT calculation in this work and most reported works are only focus on the thermodynamics, due to the complexity of experimental condition the real onset potential from LSV test are not only correlated with thermodynamic onset potential but also affected by the diffusion of oxygen molecules, the transport of electrolyte to active sites on catalysts, the surface area and so on.

For proton-transfer steps, reaction free energies are regarded as approximate values of activation barriers, since detailed calculations for the transfer of a solvated proton to adsorbed OH^- show the neglect of overbarriers has been proven as a very good approximation for a situation where the proton transfer is downhill in energy. This approximation may result in a slight overestimation of activity for a given proton-transfer elementary step, but can still qualitatively represent the right relative energetic ordering of the various proton-transfer elementary steps.

So we only focus on the activity trend (the theoretical onset potential) of these calculated models and it matches well to the experimental results, for example, the activity of synthesized Mg-N-C catalyst is comparable to that of Fe-N4-C and better than Co-N4-C reported as shown in **Table S9** in supplementary information page 52 line 748, and the results of activity trend match well with the trend of DFT prediction.

Yes, the adsorption energy of OH^* (ΔG_{OH^*}) also works well in describe the experimental results. From the calculation, we note that adsorption free energy of OH^* for most Mg, Al and Ca cofactors is exothermic and much lower than the ideal value of 1.23eV, they are too affinitive to OH species so we use the ΔG_{OH^*} to evaluate the theoretical onset potential. Based on Sabatier principle (P. Sabatier, La catalyse en chimie organique, 1920.), too weak interaction induces sluggish reaction-intermediate formation and will slow down the reaction, while if this interaction is too strong, it can block the catalysts surface and slowdown the reaction. The calculation results show that compared to the high affinity of Al and Ca cofactors the Mg cofactors are much weak and closer to the ideal value, so the Mg cofactors is predicted better activity, this prediction is confirmed by the experimental results that the synthesized Mg-N-C is much better than the Al- and Ca-based catalysts from the same method (see manuscript page 11 line 314). Therefore, the adsorption energy of *OH also works well in describe the experimental results.

Comment 4:

Personally, I am quite confused by Fig. 4i and Table S9, the $E_{1/2}$ for Mg-N-C and Pt/C in both alkaline and acidic solutions. In table S9, the $E_{1/2}$ is 910 mV and 790 mV for Mg-N-C in alkaline and acidic solutions, and both of the values are higher than that for Pt/C (860 mV in alkaline of 780 mV in acidic). Otherwise, the ORR

performance of Mg-N-C and Pt/C in alkaline is higher than that in acidic medium. However, there is an opposite conclusion in Fig. 4i?

Reply 4:

Thanks for your reminding, we are sorry to say it's our mistake. We had test the performance of our catalysts many times with independent groups and make an error bar for the performance of catalysts as shown in **Fig. 4i**, as the reference, the acidic performance of Pt/C is located around 840 ± 5 mV with catalysts loading of 0.2 mg cm^{-2} , the performance of Pt/C is better than our Mg-N-C catalysts (790 ± 10) in acidic media. In **Table S9**, we compared the performance to transition metal (Fe, Co and Mn) based catalysts. Our catalysts is better than the non Fe-based catalysts in acidic media and far exceed the carbon-based catalysts but is not higher than the commercial Pt/C. We have made a revision and given a schematics as shown in supplementary page 55 line 748 for better comparison.

Comments 5:

The calculation results used to establish a volcano map are too less in Fig 2d. For an ideal volcano should include both side location and summit of volcano map.

Reply 5:

Thanks for your suggestion.

For volcano map, it is built from the 3D volcano map as shown below, the free energy of elementary steps can be transformed to be the variables related to ΔG_{OOH^*} and ΔG_{OH^*} , and the dots located on the projection of the 3D maps. We built this 3D map with 12 calculated models from the method reported by Norskov¹¹, you can find the similar volcano map in this reference. And as you can see in Fig. 2d, it is included some dots located at the side of map.

We have added the details about DFT calculation in supplementary information page 10 line 270.

Comments 6:

The result of EIS in Pt/C was missing in Fig.S23.

Reply 6:

Thanks for the suggestion, we have added in Fig. S23 as shown below.

The charge transfer resistance of Mg-N-C is much smaller than N-C and is similar to Pt/C.

Comments 7:

The Fig. 6 should be Fig. 4.

Reply 7:

We have revised it in manuscript page 12 line 318.

Comments 8:

There are already some reports on catalytic activity of main metal group N-C catalysts that should be included (e.g., *Advanced Materials*, 29, 1606635 (2017)).

Reply 8:

We are pleased to find this reference, and it is helpful to support our conclusion in this study. Both the reported literature and our work have predicted the potential application of main group metal Mg in ORR. Thanks for your suggestion and we have added this reference number 27.

Comments 9:

The author calculated the onset potential for the catalysts and compared with Pt. how did the authors include the effect of electrolytes (alkaline and acidic).

Reply 9:

For each elementary step, the Gibbs reaction free energy ΔG is defined as the difference between free energies of the initial and final states and is given by the expression:

$$\Delta G = \Delta E + \Delta ZPE - T\Delta S + \Delta G_U + \Delta G_{pH}$$

where ΔG_{pH} is the correction of the H^+ free energy. $\Delta G_{pH} = -k_B T \ln[H^+] = pH \times k_B T \ln 10$. Hence, the equilibrium potential U_0 for four-electron transfer ORR at $pH = 14$ was determined to be 0.402V vs NHE or 1.23V vs RHE according to Nernst equation: $E = E_0 - 0.0591 pH$, $U_0(\text{RHE}) = U_0(\text{NHE}) + 0.828V = 0.402 + 0.828 = 1.23V$, where the reactant and product are at the same energy level.

The reaction free energy of (1)–(4) for ORR can be calculated using the following equations:

$$\Delta G_1 = \Delta G_{OOH^*} - 4.92$$

$$\Delta G_2 = \Delta G_{O^*} - \Delta G_{OOH^*}$$

$$\Delta G_3 = \Delta G_{OH^*} - \Delta G_{O^*}$$

$$\Delta G_4 = -\Delta G_{OH^*}$$

The onset potential is calculated by:

$$U_{\text{RHE}}^{\text{onset}} = -\max\{\Delta G_1, \Delta G_2, \Delta G_3, \Delta G_4\}$$

In this manuscript, we focus on the adsorbates strength on alkaline condition.

The free energy of elementary step in acidic condition can be transform by the Nernst equation, but it doesn't affect the trend of adsorbates strength on metal sites.

We have revised the supplementary information in page 7 line 216 and give more details about the DFT calculation.

Comments 10:

In nature, Chlorophyll contains MgN4 with Mg²⁺. But in this report, M-N2C is considered as the most active center with Mg²⁺. There may be mixed bond configuration in MgN2C.

Reply 10:

We have noticed the MgN4 configuration in Chlorophyll with Mg²⁺, and we have compared the potential activity for Mg-N4 and Mg-N2 configurations, and DFT results show that the OH* adsorption strength on Mg-N4 is stronger than Mg-N2 and the energy barrier of the second elementary step (from OOH* to O*) for Mg-N4 is too large. By rise of p-band center position of Mg-N2 compared to Mg-N4, the OH* adsorption strength is weakened, therefore the activity for Mg-N2 is better than Mg-N4.

Although DFT calculation and experimental results show that the moiety in catalyst is more likely to the Mg-N2 configuration, we can't exclude the other possible configuration like Mg-N4 existing in the Mg-N-C catalyst.

So we carefully discussed this part in manuscript page 10 line 273 as shown below:

“Thus, the results of DFT and XANES for Mg-N-C indicate that the most likely high active moiety in Mg-N-C is the MN2C cofactor (the different spatial configurations similar to MN2C are compared in Fig. S39)”

References:

1. Deng, D. *et al.* A single iron site confined in a graphene matrix for the catalytic oxidation of benzene at room temperature. *Science Advances* **1**, e1500462 (2015).
2. Chen, Y. *et al.* Isolated Single Iron Atoms Anchored on N-Doped Porous Carbon as an Efficient Electrocatalyst for the Oxygen Reduction Reaction. *Angewandte Chemie* **129**, 7041–7045 (2017).
3. Yang, Y. *et al.* O-, N-Atoms-Coordinated Mn Cofactors within a Graphene Framework as Bioinspired Oxygen Reduction Reaction Electrocatalysts. *Advanced Materials* **30**, 1801732 (2018).

4. Ingason, A. S., Eriksson, A. K., Lewin, E., Jensen, J. & Olafsson, S. Growth and structural properties of Mg:C thin films prepared by magnetron sputtering. *Thin Solid Films* **518**, 4225–4230 (2010).
5. Ramachandran, M. & Reddy, R. G. Direct reduction of magnesium oxide to magnesium using thermal plasma technology. *Mining, Metallurgy & Exploration* **32**, 30–37 (2015).
6. Petnikota, S. *et al.* Electrochemistry-related aspects of safety of graphene-based non-aqueous electrochemical supercapacitors: a case study with MgO-decorated few-layer graphene as an electrode material. *New J. Chem.* **43**, 9793–9801 (2019).
7. Zheng, Y. *et al.* Toward Design of Synergistically Active Carbon-Based Catalysts for Electrocatalytic Hydrogen Evolution. *ACS Nano* **8**, 5290–5296 (2014).
8. Zheng, Y., Jiao, Y., Jaroniec, M. & Qiao, S. Z. Advancing the Electrochemistry of the Hydrogen-Evolution Reaction through Combining Experiment and Theory. *Angewandte Chemie International Edition* **54**, 52–65 (2015).
9. Greeley, J. *et al.* Alloys of platinum and early transition metals as oxygen reduction electrocatalysts. *Nature Chemistry* **1**, 552–556 (2009).
10. Calle-Vallejo, F., Martínez, J. I., García-Lastra, J. M., Abad, E. & Koper, M. T. M. Oxygen reduction and evolution at single-metal active sites: Comparison between functionalized graphitic materials and protoporphyrins. *Surface Science* **607**, 47–53 (2013).
11. Vojvodic, A. & Nørskov, J. K. New design paradigm for heterogeneous catalysts. *Natl Sci Rev* **2**, 140–143 (2015).

[Reviewer 2]

Comments:

The article entitled “Learning from Nature, a case to turn main group element Mg into a highly active electrocatalyst for oxygen reduction reaction” deals with DFT calculation and experimental study of M-N-C type of materials for ORR.

Despite the high interest to the subject in the field this article does not contribute to the field on the exceptional level required by Springer Nature Publishing group. Material has extremely low performance in acidic media. Performance in alkaline is similar to previously published data. Synthetic approach was presented decades ago. Decision - Decline.

Reply:

Thanks for your suggestion. But we feel that you may understand our work.

Here we report a case to tune the catalytic activity of Mg-based electrocatalyst by changing the coordination environment. From comments, all reviewers evoke a great interest to our work towards main group metal electrocatalysis. Despite innumerable applications of transition metal(TM)-based catalysts, main group metal is rarely concerned in electrocatalysis. Contrary to many catalytically-active TM-catalysts, classical main-group compounds do not possess the combination of empty and filled orbitals that is crucial for the complex electronic processes involved in the elemental steps of catalytic cycles. Therefore the development of catalysts based on the main group elements thus requires the design and application of unique strategies.

In this work, we focus on the electronic structure-coordination relationship and report a case of high performance electrocatalyst based on main group metal Mg. I fully understand the stringent screening process. However, we feel that the reviewer 2 may misunderstand this work.

Despite the topic of main group electrocatalysis, as the **Table 1-2** listed below, we have compared the performance of our electrocatalyst to the most active Fe-based catalysts and non Fe-based catalysts reported on high impact journals in recent years. To be more obviously, we also make a diagram (as presented in **Figure 1**) to show the performance comparison in both alkaline electrolyte and acidic media. As you can see, our material is at the top of list. Specially, although Fe-based catalysts is active in ORR, while they are criticized for their participation in and/or promotion of the Fenton reactions which can degrade the polymer membrane in PEMFCs, so non Fe-based catalysts is desirable. Our material is comparable to the Fe-based catalysts and far exceed the non Fe-based catalysts in both acidic and alkaline electrolyte, such as Co, Mn, Cu or carbon-based catalysts.

Fig. 1. The ORR activity comparison between this work and literatures in both

alkaline and acidic electrolyte.

Table 1. The alkaline ORR activity comparison between this work and reported materials in recent years.

Reported time	Co-based electrocatalysts	Half-wave potential (V vs. RHE)	References
2015.10	Co-N-C-0.8 NPHs	0.871	ACS Catal.2015, 5, 12, 7068-7076
2016.1	ZIF-67 derived carbon	0.87	Nature Energy. 2016, 1, 15006
2016.6	CoSAs/N-C(900)	0.88	Angew. Chem. Int. Ed. 2016, 55, 10800
2017	NC@Co-NGC DSNC	0.82	Adv. Mater. 2017, 29, 31
2017.2	Co-C ₃ N ₄ /CNT	0.85	J. Am. Chem. Soc. 2017, 139, 3336–3339
2018.7	Co ₃ O ₄ /HNCP-40	0.845	ACS Catal.2018, 8, 9, 7879-7888
2018.11	N-C-CoO _x	0.84	Angew. Chem. 2019, 131, 1058–1063
2019.9	Co-N-C	0.91	Adv. Energy Mater. 2019, 9, 1900149
2019.9	Mg-N-C	0.91	This work

Reported time	Fe-based electrocatalysts	Half-wave potential (V vs. RHE)	References
2017.4	Fe-N-C	0.90	Angew. Chem. 2017, 129, 7041–7045
2018.10	CNT/PC	0.88	J. Am. Chem. Soc. 2016, 138, 45, 15046-15056
2019	CAN-Pc(Fe/Co)	0.84	Angew. Chem. Int. Ed. 10.1002/anie.201908023
2019.6	Co-Fe alloy	0.89	J. Am. Chem. Soc. 2019, 141, 27, 10744-10750
2019.9	Fe-NCNWs	0.91	ACS Catal. 2019, 9, 7, 5929-5934
2019.9	Mg-N-C	0.91	This work

Reported time	Mn or Cu-based electrocatalysts	Half-wave potential (V vs.	References
---------------	---------------------------------	----------------------------	------------

		RHE)	
2016.9	Cu-N@C	0.80	Energy Environ. Sci., 2016, 9, 3736
2018.6	Mn-N-O	0.86	Adv. Mater. 2018, 1801732
2019.5	Mn@NG	0.82	Applied Catalysis B: Environmental 257 (2019) 117930
2019.8	Cu-N-C-ICHP	0.85	Small 2019, 1902410
2019.8	Cu ISAS/NC	0.92	Nat. Commun. (2019) 10:3734
2019.9	Cu/G	0.85	Nano Energy 66 (2019) 104088
2019.9	Mg-N-C	0.91	This work

Reported time	Carbon-based electrocatalysts	Half-wave potential (V vs. RHE)	References
2016,12	N,S co-doped carbon	0.87	Adv. Mater. 2017, 29, 1604942
2017.1	1100-CNS	0.85	Energy Environ. Sci., 2017,10, 742-749
2017.7	NHPC-900-1000	0.84	ACS Catal.2017796082-6088
2018.2	N-Doped Carbons	0.69	Adv. Funct. Mater.2018, 28, 1707284
2018.10	NFLGDY-900c	0.87	Nature Chemistry volume 10, pages 924–931 (2018)
2018.10	N-HC@G-900	0.85	Angew. Chem. Int. Ed. 2018, 57, 16511
2019.9	SNBCs	0.85	ACS Catal.2019,9,4,3389-3398
2019.9	Mg-N-C	0.91	This work

Table 2. The acidic ORR activity comparison between this work and reported materials in recent years.

Reported time	Fe-based electrocatalysts	Half-wave potential (V vs. RHE)	References
2015.5	Fe-N-C nanofiber	0.62	Angew. Chem. Int. Ed. 2015, 54, 8179
2015.6	Fe-N-C	0.84	Nat. Commun. 2015, 6, 8618
2016.7	Fe-N-C	0.82	Nano Energy 25 (2016) 110–119
2017.8	(CM+PANI) Fe-C	0.80	Science, 2017, 357, 479-484
2017.11	Fe SAs/N-C	0.75	J. Am. Chem. Soc. 2017, 139, 48, 17281-17284
2018.10	CNT/PC	0.79	J. Am. Chem. Soc.2016, 138, 45, 15046-15056
2018.12	PF-2	0.771	Science 362, 1276–1281 (2018)

2019.6	Fe-N-C	0.88	Energy Environ. Sci., 2019, 12, 2548
2019.9	Fe-NCNWs	0.82	ACS Catal. 2019, 9, 7, 5929-5934
2019.9	Mg-N-C	0.79	This work

Reported time	Co, Mn or Cr-based electrocatalysts	Half-wave potential (V vs. RHE)	References
2015.10	Co-N-C-0.8 NPHs	0.761	ACS Catal. 2015, 5, 12, 7068-7076
2017.11	Co SAs/N-C	0.747	J. Am. Chem. Soc. 2017, 139, 17281–17284
2018.10	Mn-N-C-second	0.80	Nature Catalysis volume 1, pages 935–945 (2018)
2018.10	Mn-N-C	0.78	Applied Catalysis B: Environmental 243 (2019) 195–203
2019.7	Cr-N-C	0.774	Angew. Chem. Int. Ed. 2019, 58, 12469–12475
2019.9	Mg-N-C	0.79	This work

Reported time	Carbon-based electrocatalysts	Half-wave potential (V vs. RHE)	References
2015.10	N, P-doped porous carbon	0.62	Nature Nanotech. 2015, 10, 444
2016.4	N, P-doped CGHNs	0.68	Adv. Mater. 2016, 28, 4606
2017.1	1100-CNS	0.73	Energy Environ. Sci., 2017, 10, 742-749
2018.10	N-HC@G-900	0.65	Angew. Chem. Int. Ed. 2018, 57, 16511
2019.9	Mg-N-C	0.79	This work

Furthermore, the reviewer don't give further explanation for that "**Synthetic approach was presented decades ago**". Firstly, we haven't found the same or similar synthetic method to synthesize the Mg-based metal-organic-frameworks (MOFs) to use in ORR. Secondly, the common materials synthesis method is limited. In this work, the Mg-based MOFs is synthesized via solution, and main group metal based MOFs is rarely reported compared to transition-metal based MOFs. So we think the comments is unfair.

In conclusion, we think the reviewer 2 had misunderstand our work. And we have revised the manuscript according to the other two reviewers' suggestion.

In this work, we report a case about the main group metals Mg-based catalysts with high-performance in ORR which is different from the common Fe or transition metal based materials, we believe that our paper is original, significant and appealing to the readership of **Nature communication**.

[Reviewer 3]

In this manuscript, the catalytic activity of Mg-based electrocatalyst for the ORR is reported in acidic and basic electrolytes. The Mg-based electrocatalyst offers high ORR activity surpassing that of the Pt/C commercial electrocatalyst.

Comments 1:

Please revise the XPS plots shown in Fig. 3 (g) for the Mg-N-C for improving the position of the fit with the experimental data points.

Reply 1:

Thanks for your suggestion, we have revised it in manuscript page 10 line 266 as shown below:

Fig. 3g. The XPS result of the N 1s spectrum for Mg-N-C and MgPc.

Comments 2:

Please make sure that all figures are correctly numbered and labeled. Please replace figure number 6 with figure 4.

Reply 2:

We have revised it in manuscript page 12 line 318.

Comments 3:

Line number 264: Here, Fig. 5B is mentioned, while this figure number is not seen in the manuscript.

Reply 3:

The Fig. 5b is the reaction pathway of MN2C and you can find it at the top left corner of Fig. 5b in the manuscript (page 14 line 366).

Comments 4:

The reference number 10 and 16 are similar. However, their numbers are different.

Reply 4:

Thanks for your suggestion, we have revised it in our manuscript (page 16 line 417 and 429).

Comments 5:

The reference style is not according to the journal requirement, and it is not consistent.

Reply 5:

We have revised the reference style by Zetero.

Comments 6:

Why MN3C and MN4C exhibits lower ORR activity compared to MN2C?

Reply 6:

The reaction activity is strongly related to the intermediates adsorption strength. Based on Sabatier principle (P. Sabatier, La catalyse en chimie organique, 1920.), too weak interaction induces sluggish reaction-intermediate formation and will slow down the reaction, while if this interaction is too strong, it can block the catalysts surface and slowdown the reaction. From the DFT calculation, the OH* adsorption strength is too strong on MN3C to catalyze the ORR ($\Delta G_{OH^*} = -4.37\text{eV}$), while the OH* adsorption strength on MN2C ($\Delta G_{OH^*} = 0.77\text{eV}$) is near optimal value (1.23eV)

compared to other two cofactors. So there is a large energy barrier in elementary step for MN3C and MN4C compared to MN2C, as a result, their ORR activity are inferior to MN2C.

Comments 7:

Please describe why the Mg sites are active compared to Al even though both transforms O₂ to OOH radical are exothermic.

Reply 7:

We have added more description in the manuscript page 4 line 127 as shown below: “It reveals that O₂ transform to OOH radical at Mg and Al sites are exothermic but OH* adsorption strength at Al and Ca sites are too stronger than Mg sites which induces large energy barrier in elementary steps, so only Mg sites show better ORR performance.”

Sincerely,
Shuai Liu

Corresponding Author: Qianwang Chen
Hefei National Laboratory for Physical Science at Microscale and Department of
Materials Science & Engineering,
University of Science and Technology of China, Hefei 230026, China
Email: cqw@ustc.edu.cn
Fax & Tel: +86-551-63603005

Reviewers' comments:

Reviewer #1 (Remarks to the Author):

The authors have answered all the questions. The paper is publishable now.

Reviewer #3 (Remarks to the Author):

Review on NCCOMS-19-22013A-Z Learning from Nature, a case to turn main group element Mg into a highly active electrocatalyst for oxygen reduction reaction

The authors explored main group metals for oxidation reduction reaction (ORR) by manipulating the coordination numbers, and the theoretical calculation and experimental results showed that MgN₂C was the active site. The paper is nicely written, and the results are exciting. There are some details should be explained before publishing.

Comments:

1. How to confirm the Mg is incorporated into the graphene carbon matrix as illustrated in Figure 1, or it is just bonded to the edged of graphene? How would that potentially affect the results and discussion?
2. Little explanation is given on why Mg-N-C ORR activity is slightly better than that of Pt/C in alkaline condition.
3. To better quantify the activity contribution by MgN₂C, please present the percentage of inhibited activity in the KSCN poisoning test.
4. It can be observed from Fig. S22 a&b that there were two small peaks between 1302 and 1303 eV in Mg 1s, and one peak between 49.5 and 50.0 eV in Mg 2p. They seem to be corresponding to the presence of metallic Mg, but the authors seem to neglect these peaks.
5. What is the evaporation temperature of Mg? Is there any evidence to support this claim? Could the author calculate the evaporation rate by comparison to the production of N-C-no metal?
6. It is surprised to see N-C sites were little active for the ORR activity in this study, while pyridinic nitrogen was actually one of the major species in the material, which has been widely reported as an active species for the ORR activity. Could the author provide some explanation regarding the discrepancies? Please see the reference: Guo, D. H. et al. *Science* 351, 361–365 (2016). In addition, recent studies have pinpointed the importance of defects in nitrogen-doped carbon in ORR activity (*Nature Catalysis* 2, 688–695 (2019); *Nature Catalysis* 2, 642–643 (2019)). The current manuscript seems to overlook the effects of defect created before and after the introduction of Mg. Please provide some analysis.
7. To help fabricate highly active electrocatalysts in the future, how to control the synthesis of more MgN₂C in the catalysts? Please provide some comments and discussion if possible.

Response to the comments

Dear Editor and Reviewers:

Thanks for your letter and the reviewers' comments concerning our manuscript entitled "**Learning from Nature, a case to turn main group element Mg into a highly active electrocatalyst for oxygen reduction reaction**" (Manuscript ID: NCOMMS-19-22013A-Z). Those valuable and professional comments are very useful for us to improve our manuscript. We have studied all the comments carefully.

The corrections are highlighted with **blue color** in the revised main article and supplementary information.

[Reviewer 1]

Comments:

The authors have answered all the questions. The paper is publishable now.

Reply:

Thanks for your constructive advice to improve the quality of our manuscript.

[Reviewer 3]

Comments:

Review on NCCOMS-19-22013A-Z Learning from Nature, a case to turn main group element Mg into a highly active electrocatalyst for oxygen reduction reaction. The authors explored main group metals for oxidation reduction reaction (ORR) by manipulating the coordination numbers, and the theoretical calculation and experimental results showed that MgN₂C was the active site. The paper is nicely written, and the results are exciting. There are some details should be explained before publishing.

Reply:

Thanks for your valuable and positive evaluation and we had revised the manuscript according to your suggestions.

Comments 1:

How to confirm the Mg is incorporated into the graphene carbon matrix as illustrated in Figure 1, or it is just bonded to the edged of graphene? How would that potentially affect the results and discussion?

Reply 1:

Thanks for your suggestion. In this study, MN₂C moiety with high theoretical activity in ORR is well matched with XANES and EELS characterization results. Therefore, it is suggested that the MN₂C is the most preferred moiety in this catalyst.

From the HAADF-STEM images in Figure 3c (in manuscript page 9) and Figure S21 (in supplementary information page 28), we can see most Mg atoms are deposited in the carbon matrix, but we still cannot exclude that there may be a small amount of unobserved Mg atoms located at the edge of graphene. In the old version of our manuscript, we did haven't take the edge-bonded moieties into consideration, so thanks for your constructive suggestion. To consider their potential activity contribution, we had evaluated the potential activity of edge-bonded moieties as shown below.

We have built the typical Mg atom bonded armchair edge and zigzag edge moieties as shown below in Figure S40, while DFT calculation shows their theoretical ORR activities are poor compared to that of MN₂C. From the free energy diagrams in Figure S41 g and h, we can see the OH adsorption strength for the two moieties are too strong to catalyze the ORR.

So we suggested that MN₂C located at the carbon matrix is the most preferred configuration. And we had revised the manuscript adding the discussion in manuscript page 10 line 275 and in supplementary information page 47-48.

Figure S40. Configurations of edge-bonded Mg cofactors. (a) The top and side views of armchair edge-bonded MgN₂C cofactor. (b) The top and side views of zigzag edge-bonded MgN₂C cofactor.

Figure S41. The configurations of intermediates on edge-bonded Mg cofactors and corresponding free energy diagrams. (a-c) The top and side views of OOH^* , O^* , OH^* at armchair edge-bonded MgN2C cofactor. (d-f) The top and side views of OOH^* , O^* , OH^* at zigzag edge-bonded MgN2C cofactor. (g and h) The free energy diagrams of armchair edge-bonded MgN2C and zigzag edge bonded MgN2C.

Comments 2:

Little explanation is given on why Mg-N-C ORR activity is slightly better than that of Pt/C in alkaline condition.

Reply 2:

Thanks for your suggestion. As you can see the label in **Figure 4a** in manuscript, the loading of 20wt% Pt/C (0.2mg cm^{-2}) is half of Mg-N-C (0.4mg cm^{-2}). We have test the LSV curves of 20wt% Pt/C with different loading as shown below in **Figure R1**, you can find that the activity of Pt/C with higher loading (0.4mg cm^{-2}) is better. The activities of Pt/C as reference in most papers reported are around 860mV with loading of 0.2mg cm^{-2} , so as we did. And also we had made an error bar as shown in **Figure 4i** to carefully evaluate the activity of our catalyst and the Pt/C reference.

Figure R1. The LSV curves for 20wt% commercial Pt/C with different loading amounts in alkaline electrolyte.

Comments 3:

To better quantify the activity contribution by MgN₂C, please present the percentage of inhibited activity in the KSCN poisoning test.

Reply 3:

Yes, we have given the values of inhibited activity in the KSCN poisoning test. As shown in supplementary information page 37 line 595 and it is also pasted below: “after adding 0.01M KSCN, there is a negative shift of 109 mV in $E_{1/2}$, indicating loss of activity.”

Comments4:

It can be observed from Fig. S22 a&b that there were two small peaks between 1302 and 1303 eV in Mg 1s, and one peak between 49.5 and 50.0 eV in Mg 2p. They seem to be corresponding to the presence of metallic Mg, but the authors seem to neglect these peaks.

Reply 4:

Thanks for your suggestion. For the Mg-N-C catalyst, the Mg content is only 1.3wt% (about 0.6 at%), so the XPS result (the original data in Figure R2 is shown below) in Figure S22 a and b are shown with noise signal, and it is not as smooth as that of the metal oxides or metals. We had discussed with the technician of XPS for

his advice and it is now verified.

Figure R2. The original XPS data of (a) Mg 1s and (b) Mg 2p signals. The dash red area is the noise signal from the test due to the low contents of Mg in Mg-N-C.

Comments 5:

What is the evaporation temperature of Mg? Is there any evidence to support this claim? Could the author calculate the evaporation rate by comparison to the production of N-C-no metal?

Reply 5:

It is a good suggestion and we hadn't taken it into account in previous manuscript version.

The melting point and boiling point of metal Mg is 651°C and 1108°C, for nano-metal particles it could be much lower due to the high surface area.

We had synthesized the precursor of Mg-HMT in different temperatures for one hour. And we found the continuous change of pyrolysis as shown below in Figure S16. From the TGA curves of pyrolysis in Figure S16 a, there is a weight loss in temperature range 500-700°C. According to the Figure S16b, the precursor is not completely change to carbon in 500°C, until 600°C there is a broad peak at 26-28° in XRD, and meantime there is slight signals at 42.8° and 62.2°, which suggests the existence of MgO at this temperature. In temperature 700°C, the signal of MgO is stronger than that at 600°C, which reveals the MgO amount is become more. While it is weaker at 800°C and is almost absence at 900°C.

It is interesting and we have referred the references. It had been reported that the MgO can be reduced by carbon^{1,2} and especially it can occur at 900°C¹, so we think the MgO is reduced by carbon and transferred to Mg and CO¹ during pyrolysis, and we found the quartz crucible was polluted so that it turned to be gray with metallic luster after heat treatment at 900°C. Although XRD patterns didn't give an apparently peak of MgO in pyrolysis at 900°C before acidic etching, FT-IR and XPS still showed a weak signal of lattice oxygen referred to Mg-O bond as shown in Figure S19 and Figure S22 d, so these results confirm the reduction of MgO by carbon and a little amount MgO undetected still existence after the pyrolysis at 900°C. It also indicates

that the Mg cofactors probably be formed at around temperature 900°C in the reduction process. There are some papers verifying this similar transformation process, like FeO_x is transferred to Fe-N coordination in the pyrolysis³⁻⁵.

We had added this discussion in manuscript page 8 line 222 and in supplementary information page 25 line 467.

Fig. S16. (a) Thermogravimetry curve of Mg-HMT precursor and (b) corresponding XRD patterns.

Comments 6:

It is surprised to see N-C sites were little active for the ORR activity in this study, while pyridinic nitrogen was actually one of the major species in the material, which has been widely reported as an active species for the ORR activity. Could the author provide some explanation regarding the discrepancies? Please see the reference: Guo, D. H. et al. *Science* 351, 361–365 (2016). In addition, recent studies have pinpointed the importance of defects in nitrogen-doped carbon in ORR activity (*Nature Catalysis* 2, 688–695 (2019); *Nature Catalysis* 2, 642–643 (2019)). The current manuscript seems to overlook the effects of defect created before and after the introduction of Mg. Please provide some analysis.

Reply 6:

Thanks for your suggestion, we had added the discussion about contribution of carbon defect in the new manuscript version.

We had carefully read the reference paper about the almost pure pyridinic-nitrogen doped graphene⁶. The half-wave potential ($E_{1/2}$) of this material (N-GNS-3) is about 0.6V (vs. RHE) at current density of 1mA cm^{-2} in acidic condition, the N-C reference we used is the various type of nitrogen atoms doped graphene (it is not the pure pyridinic-nitrogen) with $E_{1/2}$ of 480mV at 1.6 mA cm^{-2} , so we think it's common that the N-C reference we used is not active in acidic condition.

Yes, we had noticed the recent study reported the intrinsic defect especially the form of pentagon edge sites by nitrogen doping⁷, and they pointed that it was pentagon edge sites that active in acidic activity not nitrogen doping sites. So based on current characterization technologies, it is so difficult to identify the real active

sites for carbon-based materials.

But one thing should not be neglected is that there could be the activity contribution of carbon sites in Mg-N-C. As you can see in Figure S31 in page 36 and line 596 in supplementary information, after KSCN poisoning, the activity of Mg-N-C has a negative shift by 109mV, while it still has an $E_{1/2}$ of more than 610mV. This result confirms the activity contribution of metal Mg sites and also infers the activity contribution of carbon sites. Our DFT calculation also had revealed the dual sites (Mg and the C1 sites) reaction mechanism as discussed in manuscript page 13 line 349.

Moreover, we had noticed that most carbon-based catalysts are not as active as metal-based catalysts and we had summarized and compared the activity of different materials reported in high impact journals as shown in Table S9 in page 57 in supplementary information. And the activity of Mg-N-C far exceeds the carbon-based catalysts. So we think the Mg site and neighbor-carbon site is the most preferred active sites in Mg-N-C catalyst.

We had added this discussion as shown below and cited these meaningful references as references 24, 25.

Revised at page 36 and line 596 in supplementary information as pasted below:

“While it still has an $E_{1/2}$ of more than 610mV. This result confirms the activity contribution of metal Mg sites but also inferred the activity contribution of carbon sites.^{24,25}”

Comments 7:

To help fabricate highly active electrocatalysts in the future, how to control the synthesis of more MgN₂C in the catalysts? Please provide some comments and discussion if possible.

Reply 7:

It is a good question and we are doing this job currently. We have found the method to synthesize more active catalysts in large amount, one thing I can tell is that we had synthesized the Mg-N-C catalysts more than 200 times in this work and one can repeat our experiment. The tip is to prolong the pyrolysis time. More information would be given in our following works.

Again, thanks for your constructive suggestion and valuable evaluation to improve the quality of our manuscript.

References

1. Jansson, S., Brabie, V. & Jönsson, P. Corrosion Mechanism of Commercial MgO–C Refractories in Contact with Different Gas Atmospheres. *ISIJ Int.* **48**, 760–767

- (2008).
2. Fruehan, R. J. & Martonik, L. J. The Rate of reduction of MgO by carbon. *MTB* **7**, 537–542 (1976).
 3. Li, J. *et al.* Thermally Driven Structure and Performance Evolution of Atomically Dispersed FeN₄ Sites for Oxygen Reduction. *Angewandte Chemie International Edition* **n/a**, (2019).
 4. Liu, Q., Liu, X., Zheng, L. & Shui, J. The Solid-Phase Synthesis of an Fe-N-C Electrocatalyst for High-Power Proton-Exchange Membrane Fuel Cells. *Angew. Chem. Int. Ed.* **57**, 1204–1208 (2018).
 5. Yang, W. *et al.* Facile Synthesis of Fe/N/S-Doped Carbon Tubes as High-Performance Cathode and Anode for Microbial Fuel Cells. *ChemCatChem* **n/a**, (2019).
 6. Guo, D. *et al.* Active sites of nitrogen-doped carbon materials for oxygen reduction reaction clarified using model catalysts. *Science* **351**, 361–365 (2016).
 7. Jia, Y. *et al.* Identification of active sites for acidic oxygen reduction on carbon catalysts with and without nitrogen doping. *Nature Catalysis* **2**, 688–695 (2019).

Sincerely,
Shuai Liu

Corresponding Author: Qianwang Chen
Hefei National Laboratory for Physical Science at Microscale and Department of
Materials Science & Engineering,
University of Science and Technology of China, Hefei 230026, China
Email: cqw@ustc.edu.cn
Fax & Tel: +86-551-63603005

REVIEWERS' COMMENTS:

Reviewer #3 (Remarks to the Author):

I am satisfied with the revised MS.

Response to the comments

Dear Editor and Reviewers:

Thanks for your letter and the reviewers' comments concerning our manuscript entitled "**Learning from Nature, a case to turn main group element Mg into a highly active electrocatalyst for oxygen reduction reaction**" (Manuscript ID: NCOMMS-19-22013B). Those valuable and professional comments are very useful for us to improve our manuscript.

[Reviewer 3]

Comments:

I am satisfied with the revised MS.

Reply:

Thanks very much for your constructive advice to improve the quality of our manuscript.

Sincerely,
Shuai Liu

Corresponding Author: Qianwang Chen
Hefei National Laboratory for Physical Science at Microscale and Department of
Materials Science & Engineering,
University of Science and Technology of China, Hefei 230026, China
Email: cqw@ustc.edu.cn
Fax & Tel: +86-551-63603005